# Trajectory World Models for Heterogeneous Environments

Shaofeng Yin [* 1]  Jialong Wu [* 1]  Siqiao Huang [1]  Xingjian Su [1]  Xu He [2]  Jianye Hao [2]  Mingsheng Long [1]

## Abstract

Heterogeneity in sensors and actuators across environments poses a significant challenge to building large-scale pre-trained world models on top of this low-dimensional sensor information. In this work, we explore pre-training world models for heterogeneous environments by addressing key transfer barriers in both data diversity and model flexibility. We introduce UniTraj, a unified dataset comprising over one million trajectories from 80 environments, designed to scale data while preserving critical diversity. Additionally, we propose TrajWorld, a novel architecture capable of flexibly handling varying sensor and actuator information and capturing environment dynamics in-context. Pre-training TrajWorld on UniTraj yields substantial gains in transition prediction, achieves a new state-of-the-art for off-policy evaluation, and also delivers superior online performance of model predictive control. To the best of our knowledge, this work, for the first time, demonstrates the transfer benefits of world models across heterogeneous and complex control environments. Code and data are available at https://github.com/thuml/TrajWorld.

## 1. Introduction

World models (Ha & Schmidhuber, 2018; LeCun, 2022) have made remarkable progress in addressing sequential decision-making problems (Hafner et al., 2020; Schrittwieser et al., 2020; Hansen et al., 2024). Trained on trajectory data, these models can simulate environments and are leveraged to either evaluate complex actions (Chua et al., 2018; Ebert et al., 2018; Tian et al., 2023) or optimize policies (Janner et al., 2019; Kurutach et al., 2018). However,

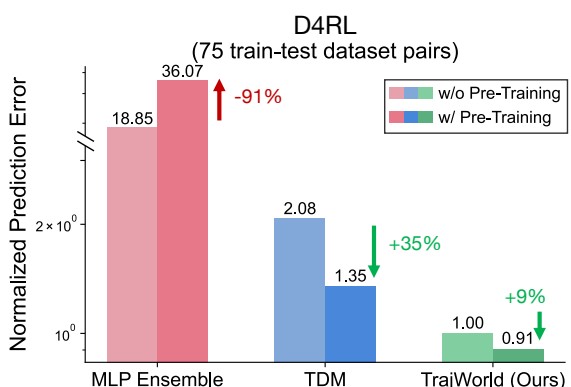

*Figure 1.* Aggregated transition prediction error (MAE) across 75 train-test dataset pairs, comparing MLP Ensemble (Chua et al., 2018), TDM (Schubert et al., 2023), and proposed TrajWorld, with and without pre-training on UniTraj dataset. Y-axis at log scale.

existing methods often learn world models *tabula rasa*, relying on data from a single, specific environment. This limits their ability to generalize to out-of-distribution transitions, demanding a substantial number of costly interactions with the environment.

In recent years, machine learning has been revolutionized by foundation models pre-trained on large-scale, diverse data (Achiam et al., 2023; Oquab et al., 2024; Kirillov et al., 2023). General world models have also been realized through pre-training, enabled by the *homogeneity* present within massive and diverse datasets of specific modalities, such as text (Wang et al., 2024b; Gu et al., 2024; Chae et al., 2025; Wu et al., 2025), images (Zhou et al., 2024), and videos (Seo et al., 2022; Wu et al., 2024a; Bruce et al., 2024; Wu et al., 2024b; Agarwal et al., 2025). However, a unique challenge of world models from Internet AI is commonly overlooked or circumvented: the *heterogeneity* inherent in sensor and actuator information, which means proprioceptive data, such as joint positions and velocities, as well as optional target positions, vary significantly across environments. Failing to properly address this heterogeneity can result in no transfer or even negative transfer.

We argue that no modality in world models should be left behind, including essential sensor information represented as low-dimensional vectors. In this work, we take a first step

---

[*]Equal contribution  [1]Tsinghua University.  [2]Huawei Noah's Ark Lab.

Shaofeng Yin <ysf22@mails.tsinghua.edu.cn>. Jialong Wu <wujialong0229@gmail.com>. Correspondence to: Mingsheng Long <mingsheng@tsinghua.edu.cn>.

*Proceedings of the 42nd International Conference on Machine Learning*, Vancouver, Canada. PMLR 267, 2025. Copyright 2025 by the author(s).

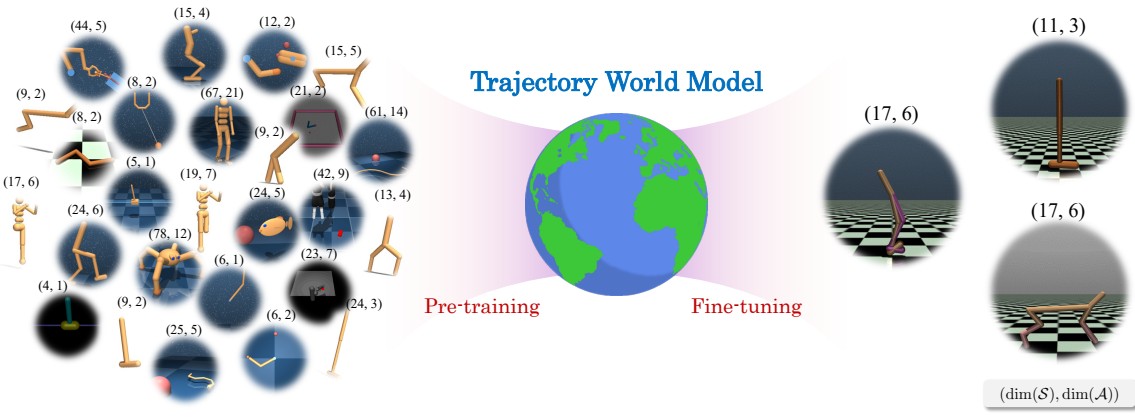

*Figure 2.* Illustration of pre-training a world model from heterogeneous environments, with each environment labeled by its state and action dimensions. A Trajectory World Model, designed for flexibility in handling divergent state and action definitions, demonstrates effective positive transfer across distinct, heterogeneous, and complex control environments.

to bridge this gap by exploring the potential of pre-training a world model to extract shared knowledge from trajectories across heterogeneous environments (illustrated in Figure 2). To this end, it is essential to overcome the transfer barriers from both data and model architecture perspectives.

**Scaling data.** To achieve strong generalization through pre-training, access to vast and diverse data is essential (Team et al., 2021). While scaling data is straightforward, the real challenge lies in scaling data while preserving diversity. Diversity in our work has two key aspects. First, it refers to the data sources, i.e., the environments from which the data is collected. Second, it concerns the data properties, specifically the distribution of the data itself. Even within the same environment, different policies at various levels can produce significantly different data distributions. To tackle these challenges, we curate the UniTraj dataset, including over one million trajectories collected from various distributions from 80 heterogeneous environments. By scaling data while maintaining these diversities, we ensure that the model focuses on the core knowledge shared across environments, thereby enabling successful transferability.

**Flexible architecture.** Previous approaches often address size variations in state and action spaces by applying zero-padding to match a maximum length (Yu et al., 2020a; Hansen et al., 2024) or employing separate input and output heads for each environment (Wang et al., 2024a; D'Eramo et al., 2020). However, zero-padding imposes a dimension limit and adds training overheads, while the separate head approach requires training new heads for new environments, hindering zero-shot transfer. A truly capable model for heterogeneous environments requires a more flexible architecture. To address this, we propose the Trajectory World Model (TrajWorld), a novel architecture that integrates interleaved variate and temporal attention mechanisms. It

is enabled to naturally accommodate varying numbers of sensors and actuators through variate attention and, more importantly, to capture their relationships in-context through temporal attention. This in-context learning capability goes beyond learning specific environment dynamics and thus enhances the model's generalizability across environments.

By pre-training our flexible TrajWorld architecture on the diverse and massive UniTraj dataset, we demonstrate, for the first time, the transfer benefits of world models across heterogeneous and complex control environments. Fine-tuning TrajWorld on 15 datasets from three previously unseen environments (Fu et al., 2020) significantly reduces transition prediction errors for both in-distribution and out-of-distribution actions (as shown in Figure 1). This improved predictive accuracy also translates to our state-of-the-art performance on off-policy evaluation (OPE) tasks (Fu et al., 2021), enabling the offline evaluation and selection of a set of complex policies for best performance. Furthermore, it also manifests in superior online performance with model predictive control (MPC).

The main contributions can be summarized as follows:

- We investigate an under-explored world model pre-training paradigm across heterogeneous environments.
- We curate UniTraj, a unified trajectory dataset, enabling large-scale pre-training of world models.
- We propose TrajWorld, a novel architecture to facilitate transfer between heterogeneous environments.
- For the first time, our experiments demonstrate positive world model transfer across diverse and complex environments, resulting in simultaneous and significant improvements in transition prediction, off-policy policy, and model predictive control.

*Table 1.* Statistics for six components of the UniTraj dataset. The checkmark (✓) represents a dataset collected or curated by ourselves.

| | Dataset | #Env. | #Episodes | #Steps | State dim. | Action dim. | Characteristic |
|---|---|---|---|---|---|---|---|
| | ExORL (Yarats et al., 2022) | 4 | 541,336 | 330,985,000 | $5 \sim 78$ | $1 \sim 12$ | Exploratory |
| | RL Unplugged (Gulcehre et al., 2020) | 7 | 5,841 | 5,841,000 | $5 \sim 67$ | $1 \sim 21$ | Experience replay |
| | JAT (Gallouédec et al., 2024) | 5 | 50,000 | 26,238,954 | $4 \sim 23$ | $1 \sim 7$ | Expert |
| | DB-1 (Wen et al., 2022) | 58 | 290 | 19,320 | $5 \sim 67$ | $1 \sim 21$ | Expert; Diversity |
| | TD-MPC2 (Hansen et al., 2024) | 30 | 672,000 | 336,000,000 | $3 \sim 24$ | $1 \sim 6$ | Experience replay |
| ✓ | Modular RL (Huang et al., 2020) | 20 | 37,199 | 19,996,902 | $7 \sim 23$ | $1 \sim 6$ | Experience replay |
| ✓ | **UniTraj** (Ours) | 80 | 1,306,666 | 719,081,176 | $3 \sim 78$ | $1 \sim 21$ | **Omnifarious** |

## 2. Problem Formulation

An environment is typically described by a Markov decision process (MDP) $\mathcal{M} = \{\mathcal{S}, \mathcal{A}, P, r, \mu\}$, specified by the state space $\mathcal{S}$ (of sensors), the action space $\mathcal{A}$ (of actuators), the transition function $P : \mathcal{S} \times \mathcal{A} \rightarrow \Delta(\mathcal{S})$, the reward function $r : \mathcal{S} \times \mathcal{A} \rightarrow \mathbb{R}$, and the initial state distribution $\mu \in \Delta(\mathcal{S})$.

Given an MDP, a trajectory of length $T$:

$$\tau = (s_0, a_0, r_1, s_1, \cdots, a_{T-2}, r_{T-1}, s_{T-1}), \quad (1)$$

is recorded as interactions between the environment and an agent, according the following protocol: starting from an initial state $s_0 \sim \mu$, at each discrete time step $t = 0, 1, \ldots$, the agent performs an action $a_t \in \mathcal{A}$ according to its policy, receives an immediate reward $r_{t+1} = r(s_t, a_t)$, and observes the next state after transition $s_{t+1} \sim P(s_t, a_t)$.

A world model $p_\theta(s_{t+1}, r_{t+1}|s_t, a_t)$, or more generally $p_\theta(s_{t+1}, r_{t+1}|s_{1:t}, a_{1:t})$, learns its parameter $\theta$ from a dataset of recorded trajectories $\mathcal{D} = \{\tau_i\}$ to approximate the underlying transition probability and reward function, thus serving as an alternative of the environment.

**Our work.** While most literature learns a world model on target environment $\mathcal{M}^t$ from scratch, we investigate an under-explored paradigm of pre-training a world model from a family of heterogeneous[1] environments $\{\mathcal{M}^1, \mathcal{M}^2, \ldots, \mathcal{M}^K\}$. Through learning from mixed trajectory data $\{\mathcal{D}^1, \mathcal{D}^2, \ldots, \mathcal{D}^K\}$, we obtain a good starting-point of the model $\theta_0$, ready for either zero-shot generalization to unseen $\mathcal{M}^t$ or fine-tuning to obtain a world model of $\mathcal{M}^t$ with strong generalization given limited data. We elaborate on the intuition behind this paradigm in Section 4.1.

## 3. UniTraj Dataset

We introduce UniTraj, a large-scale unified trajectory dataset from heterogeneous environments, to support the pre-training of a trajectory world model. To ensure diversity, we merge five publicly available datasets with different charac-

teristics. To further enhance diversity, we also by ourselves collect the training buffer of agents from a set of diverse morphologies (Huang et al., 2020). As a result, UniTraj occupies a total of 1.3M trajectories (or 719M steps) from 80 distinct environments, as summarized in Table 1. A detailed list of dataset information can be found in Appendix A.

Beyond its unprecedented scale, the collected UniTraj represents diversity in several aspects:

**Environment diversity.** UniTraj encompasses a wide range of control environments. These include not only widely-used environments from the DeepMind Control Suite (DMC) (Tassa et al., 2018) and OpenAI Gym (Brockman, 2016), but also customized embodiments and tasks proposed in Modular RL and TD-MPC2. Notably, we purposely exclude all trajectories from the HalfCheetah, Hopper, and Walker2D environment of OpenAI Gym, which are held out as our downstream test environments.

**Distribution diversity.** The dataset contains data collected from various distributions, resulting from different collection methods and policies. Specifically, data from RL Unplugged, TD-MPC2, and Modular RL are gathered by recording the training agent's replay buffer, while JAT and DB-1 data are collected through expert policies rollouts. Additionally, ExORL data are collected by storing the transitions from running unsupervised exploration algorithms (Laskin et al., 2021). The policies cover a range of approaches, including a wide range of reinforcement learning algorithms (e.g., D4PG (Barth-Maron et al., 2018), PPO (Schulman et al., 2017)) and state-of-the-art model predictive control algorithms, TD-MPC2.

By scaling up the dataset while preserving diversity, we empower the model with the potential to generalize across varied environments.

## 4. Trajectory World Models

In this section, we first explain the intuition behind the proposed Trajectory World Models (TrajWorld) (Section 4.1), then provide a detailed overview of the architecture implementation (Section 4.2), and conclude with a discussion of the pre-training and fine-tuning paradigm (Section 4.3).

---

[1]We use the term "heterogeneous" to highlight that different environments not only feature varying transition and reward functions but also, more challengingly, possess distinct state and action spaces tied to unique sets of sensors and actuators.

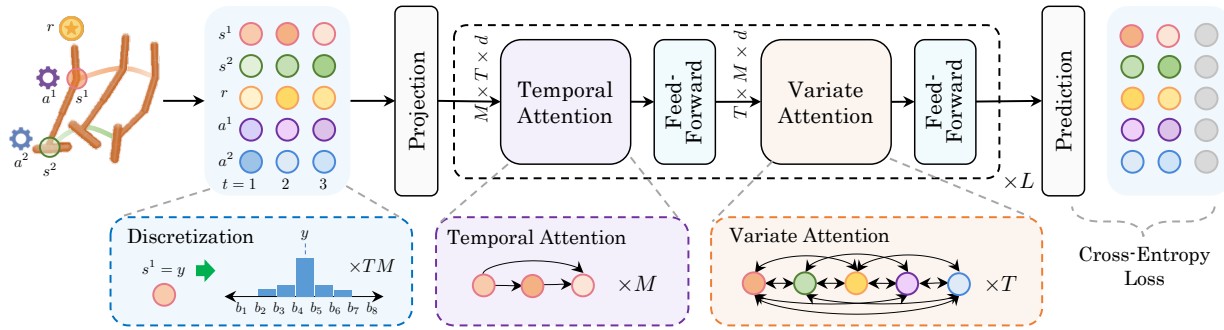

*Figure 3.* Architecture of Trajectory World Models. A trajectory is first flattened into scalars, organized into two dimensions by timesteps and variates (each variate corresponds to a single dimension in the state, action, and reward), and then discretized into categorical representations. A Transformer with interleaved temporal and variate attentions processes the inputs to predict the categorical distribution for the next timestep autoregressively. Layer normalizations and residual connections are omitted for simplicity.

### 4.1. Intuition

To address the challenges of heterogeneity and promote knowledge transfer, we make three key observations:

**Rediscovering homogeneity in scalars.** While heterogeneity often arises in differently sized vector information, there exists an inherent homogeneity at the scalar level. Each variate—a single scalar dimension in the state, action, or reward—represents a fundamental quantity with its own physical meanings of the environment, e.g., position or torque of a single joint, and can be consistently modeled, regardless of the shape of the whole vector information. This insight leads to our design choice: *Instead of treating vector information as a whole, we break it into the scalar level for processing and prediction.*

**Identifying environment through historical context.** Unlike single-environment scenarios with fixed state and action definitions, in our setting, variants can represent different quantities across environments despite the same index in the vector. While environment IDs are typically included as inputs to distinguish environments, we instead leverage the in-context learning ability of Transformers (Brown et al., 2020): history transitions can provide the context needed for the model to infer relationships between variants. This makes pre-training even more critical. By exposing the model to diverse data across environments, we encourage it to learn "how to learn environment dynamics,"—a more generalizable knowledge—rather than solely focusing on specific environments. This ability is demonstrated in Section 5.1, where our pre-trained model has satisfactory zero-shot performance. In summary, *we provide historical context instead of environment identities, guiding the model to learn to infer dynamics through context.*

**Inductive bias for two-dimensional representations.** So far, our modeling for heterogeneous dynamics involves two dimensions: one focuses on capturing the relationships among variants, and the other models how actions drive

transitions from the current state to the next. Instead of using simple one-dimensional attention over flattened sequences, explicitly modeling these two dimensions has the potential to enhance transferability in downstream tasks, as it guides the model to learn in a more structured and systematic manner. This is supported by empirical results in Section 5.2. In short, *we use a two-way attention mechanism instead of one-dimensional attention on sequences.*

### 4.2. Architecture

Building on the above intuitions, we realize a Transformer-based architecture for TrajWorld (see Figure 3).

**Scalarization.** To exploit the inherent homogeneity at the scalar level, we flatten a trajectory $\tau$ (Equation (1)) from the spaces $\mathcal{S} \subset \mathbb{R}^m, \mathcal{A} \subset \mathbb{R}^n$ into a two-dimensional representation organized by timesteps and variates:

$$X = \begin{pmatrix} s_0^{(1)} & \cdots & s_0^{(m)} & r_0 & a_0^{(1)} & \cdots & a_0^{(n)} \\ \vdots & \ddots & \vdots & \vdots & \vdots & \ddots & \vdots \\ s_{T-1}^{(1)} & \cdots & s_{T-1}^{(m)} & r_{T-1} & a_{T-1}^{(1)} & \cdots & a_{T-1}^{(n)} \end{pmatrix},$$

$$(2)$$

where $s_t^{(i)}$ denotes the $i$-th dimension of $s_t$. Padding is applied to $r_0$ and $a_{T-1}$ as zeros. This transformation converts heterogeneous trajectories of varying lengths and dimensions into matrices $X \in \mathbb{R}^{T \times M}$, where $M = m + n + 1$, which can be flexibly processed by the attention mechanism.

**Discretization and embeddings.** Transformers excel in processing discrete inputs, so we further convert scalars into categorical representations. For each variate $s^{(i)}$ or $a^{(i)}$, we define $B$ uniform bins with boundaries $b_0 < b_1 < \cdots < b_B$, where $b_0$ and $b_B$ represent the minimum and maximum values of the variate in the training data. Scalars are then mapped to these bins using one-hot encoding or Gaussian histograms (Imani & White, 2018; Farebrother et al., 2024). The resulting discrete representation $Q \in [0, 1]^{T \times M \times B}$ is

linearly projected to match the Transformer's hidden size $d$. Additionally, we apply three learned embeddings—*timestep-embedding* (TE), *variate embedding* (VE), and *prediction embedding* (PE)—to capture timestep indices, variate identities, and whether a variate is a target for prediction. Formally, for each $i \in [T]$ and $j \in [M]$:

$$Z_{ij}^0 = W_{\text{in}} Q_{ij} + \text{TE}(i) + \text{VE}(j) + \text{PE}(\mathbf{1}[j \le m+1]). \quad (3)$$

**Interleaved temporal-variate attentions.** The input $Z^0 \in \mathbb{R}^{T \times M \times d}$ is processed through a series of $L$ transformer blocks, adapted for the two-dimensional input structure. In each block $l = 1, \ldots, L$, we first apply *temporal attention*, processing each variate independently:

$$U_{1:T,j}^l = \text{CausalAttention}(Z_{1:T,j}^{l-1}), \quad \forall j \in [M], \quad (4)$$

followed by a feedforward network (FFN): $\hat{U}^l = \text{FFN}(U^l)$. Afterwards, *variate attention* is applied at each timestep:

$$V_{i,1:M}^l = \text{Attention}(\hat{U}_{i,1:M}^l), \quad \forall i \in [T]. \quad (5)$$

Since there are no causal dependencies between variates at the same timestep, no causal mask is applied during variate attention. Finally, another FFN is applied: $Z^l = \text{FFN}(V^l)$.

Through interleaved temporal and variate attentions, each entry in our model efficiently aggregates information from all variates across all previous timesteps. As previously discussed, this enables the model to infer environment dynamics in-context for transition prediction.

**Prediction and objective.** A linear prediction head, followed by a softmax operation, produces the prediction distribution $P = \text{Softmax}(W_{\text{out}} Z^L) \in [0,1]^{T \times M \times B}$. Our model is trained using a next-step prediction objective to match the categorical representation of the inputs:

$$\mathcal{L}(P, Q) = -\sum_{i=1}^{T-1} \sum_{j=1}^{m+1} \sum_{k=1}^{B} Q_{i+1,j,k} \log P_{i,j,k}. \quad (6)$$

During inference, the next-step prediction can be obtained by sampling from or taking the expectation of the predicted categorical distribution over bin centers.

### 4.3. Towards a General Trajectory World Model

We pre-train a general Trajectory World Model on offline datasets from diverse environments. This same pre-trained model can then be applied to all downstream tasks for fine-tuning. Thanks to the Transformer's flexible architecture design and in-context learning capabilities, the pre-trained knowledge becomes more transferable, benefiting a wide range of heterogeneous and complex control environments.

| Method | Prediction Error (MSE) | |
|---|---|---|
| | Pendulum | Walker2D |
| Last-step Mirroring | $1.3 \times 10^{-3}$ | $1.7 \times 10^{-2}$ |
| TrajWorld (w/o history) | $1.7 \times 10^{-5}$ | $2.1 \times 10^{-3}$ |
| TrajWorld (w/ history) | $2.9 \times 10^{-6}$ | $7.2 \times 10^{-4}$ |

(a) Environment parameters transfer.

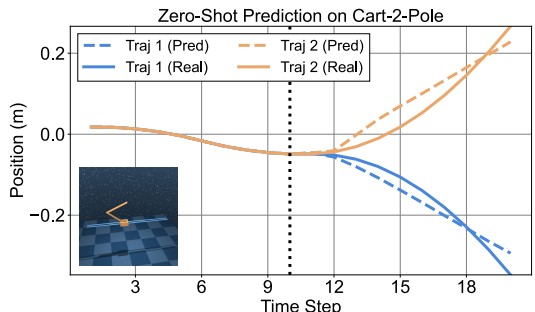

(b) Cross-environment transfer.

*Figure 4.* Zero-shot generalization. (a) Mean squared error of zero-shot transition predictions in modified Gym Pendulum (holdout gravity) and Walker2D (holdout friction etc.). (b) TrajWorld's zero-shot predictions for two Cart-2-Pole trajectories, which share 10 context steps but diverge due to differing subsequent actions.

## 5. Experiments

In this section, we test the following hypotheses:

- Large-scale trajectory pre-training can generalize effectively and even enable zero-shot generalization, contrary to the common belief (Section 5.1).
- TrajWorld outperforms alternative architectures for transition prediction when transferring dynamics knowledge to new environments (Section 5.2).
- TrajWorld leverages the general dynamics knowledge acquired from pre-training to improve performance in downstream tasks (Section 5.3).

### 5.1. Zero-shot Generalization

We first demonstrate that through in-context learning ability, TrajWorld exhibits favorable generalization across heterogeneous environments, which differ not only in their transition dynamics but also in state and action spaces.

**Environment parameter transfer.** We pre-train a TrajWorld model on data from Gym Pendulum environments with varying gravity values and evaluate its transition prediction error on holdout gravity values. As shown in Table 4a, TrajWorld achieves significantly lower prediction error in zero-shot settings compared to a naive baseline that simply mimics the last timestep. Moreover, the performance of TrajWorld deteriorates noticeably when historical information is excluded, highlighting the critical role of contexts for

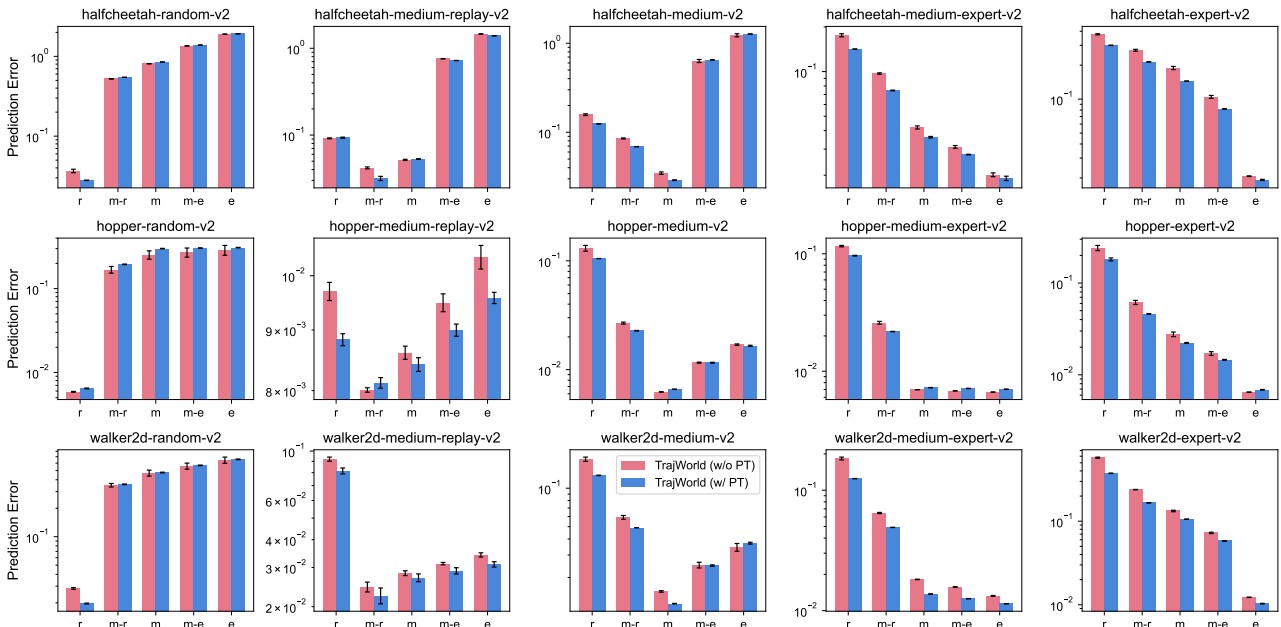

*Figure 5.* Mean absolute errors (MAE) of transition prediction for TrajWorld, with and without pre-training (PT), across different train-test dataset pairs. Each subplot corresponds to a distinct training dataset, with the test datasets shown on the x-axis (r=random, m-r=medium-replay, m=medium, m-e=medium-expert, e=expert). Error bars represent the standard deviation across three random seeds.

the model to effectively infer environment parameters. The results are consistent in a similar experiment conducted on Gym Walker2D, where friction, mass, etc., are varied.

**Cross-environment transfer.** We further find that TrajWorld, when trained on the large-scale UniTraj dataset, is also capable of zero-shot generalizing to unseen environments, Cart-2-Pole and Cart-3-Pole from DMC (Figure 4b and 13). Specifically, TrajWorld successfully infers the influence of the action value (pushing force) on the state dimension (cart position) and accurately predicts the outcomes for different action sequences performed subsequently.

## 5.2. Transition Prediction

We then evaluate how different world models benefit from pre-training for transition prediction, particularly for out-of-distribution queries, when fine-tuned to more complex, standard environments.

**Setup.** We use datasets of three environments—HalfCheetah, Hopper, and Walker2D—from D4RL (Fu et al., 2020) as our testbed. Each environment in D4RL is provided with five datasets of different distributions from policies of varying performance levels. We train world models in each of the fifteen datasets and test prediction errors of states and rewards across all five datasets under the same environment, resulting in 75 train-test dataset pairs.

**Baselines.** We compare our approach against two baselines: an ensemble of MLPs (Chua et al., 2018), widely

adopted for dynamics modeling, and TDM (Schubert et al., 2023), which is similar to our model but flattens inputs and uses one-dimensional attention. Each baseline is evaluated both for training from scratch and fine-tuning pre-trained ones on the same UniTraj dataset as TrajWorld. To enable pre-training, we pad the state and action vectors with zeros to match the same dimensionality for MLP. Additionally, we compare with our model trained from scratch.

**Results.** Figure 1 presents the aggregated mean absolute error of 75 train-test dataset pairs for various models. TrajWorld outperforms all baselines, highlighting the effectiveness of its pre-training strategy and architecture design. Notably, MLP Ensemble with pre-training performs worse than its non-pre-trained counterpart, emphasizing the importance of careful model design for world modeling across heterogeneous environments. While TDM also benefits significantly from pre-training, it still lags behind TrajWorld. This is likely because TDM naively treats everything as a 1D sequence, neglecting the unique problem structures. In contrast, TrajWorld explicitly models variate relationships and temporal transitions, leveraging different facets of dynamics knowledge from the pre-training. Moreover, TDM predicts variants sequentially, which may accumulate errors and lead to less accurate results, whereas TrajWorld predicts all variables jointly, mitigating compounding errors.

In Figure 5, we further show detailed prediction error results for TrajWorld compared to its non-pre-trained counterparts. In 12 out of 15 training datasets, fine-tuned Tra-

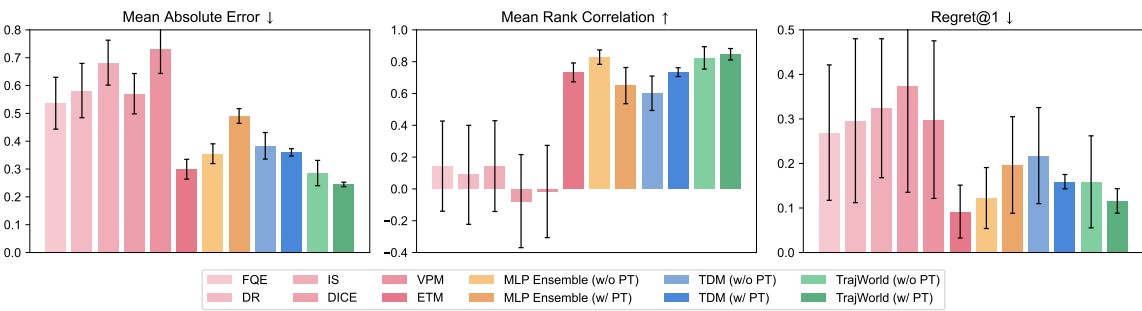

*Figure 6.* Overall off-policy evaluation (OPE) results across 15 datasets of 3 environments, averaged across three random seeds.

jWorld achieves a lower average prediction error across 5 test datasets, further validating the effectiveness of pre-training. Moreover, the transfer benefits are evident in both in-distribution and out-of-distribution scenarios, indicating that the model generalizes well even when trained and tested on transitions collected from different policies.

## 5.3. Off-Policy Evaluation

Off-policy evaluation (OPE) estimates the value of a target policy using an offline transition dataset collected by a separate behavior policy. It is commonly used to select the most performant policy from a set of candidates when online evaluation is too costly to be practical. This task provides an ideal evaluation scenario for world models, as value estimation can be acquired by rolling out the target policy within the learned world model. This is particularly advantageous for evaluating long-horizon predictions, where direct environment interaction is infeasible and model accuracy over extended timeframes is critical.

**Setup.** We adopt the DOPE benchmark (Fu et al., 2021) over various D4RL environments. The tasks in this benchmark are particularly challenging, as the target policies are of different levels and may differ significantly from the behavior policy. To perform well on these tasks, the world model must generalize well across all possible state-action distributions. Evaluation metrics include *mean absolute error* comparing estimated vs. ground-truth policy values, *rank correlation* between estimated and actual policy rankings, and *Regret@1* measuring accuracy in selecting the best policy, as detailed in Appendix B.4.3.

**Baselines.** In addition to the MLP Ensemble and TDM models mentioned earlier, we compare our approach against several other baselines. Notably, Energy-based Transition Models (ETM) (Chen et al., 2024) currently sets the state-of-the-art on this benchmark, outperforming prior methods by a significant margin. We also include the classical methods from the original DOPE paper (Fu et al., 2021) for a more comprehensive comparison.

**Results.** Figure 6 shows that TrajWorld significantly improves OPE compared to its non-pre-trained variant and outperforms all baselines in both average normalized absolute error and rank correlation. TrajWorld slightly underperforms on Regret@1, likely due to bounded reward prediction (see discussion in Appendix D). Consistent with Section 5.2, MLP Ensemble with pre-training suffers from negative transfer, showing a notable drop in performance compared to the non-pre-trained model. Although TDM also benefits from pre-training, it does not reach the same level of performance as TrajWorld. We attribute this to the same reason discussed in Section 5.2.

## 5.4. Model Predictive Control

Model predictive control (MPC) selects actions by optimizing predicted future rewards over a finite horizon using a learned world model, making it well-suited for evaluating world model performance in online control settings.

**Setup.** We evaluate MPC performance in a practical scenario where world models trained on medium-replay datasets are used to enhance medium-level proposal policies through model predictive control. Specifically, we utilize three medium-replay datasets from D4RL and medium-level policies from DOPE. Additionally, we experiment with MPC using a random shooting planner. Implementation details are provided in Appendix B.5.

**Baselines.** As in the previous section, we compare our method against the MLP Ensemble and TDM baselines, evaluting both both from-scratch and fine-tuned variants.

**Results.** Figure 7 presents results for MPC with proposal policies. Overall, MPC using TrajWorld yields the highest-performing agents, outperforming both baseline models and its from-scratch counterpart. We find that MPC leads to significant gains in the Hopper and Walker2D environments, but has limited effects in HalfCheetah, likely due to its inherent stability and lower risk of failure. In contrast, Hopper and Walker2D are fragile, and our world models help prevent unsafe actions, leading to better planning. Notably,

*Figure 7.* Model predictive control (MPC) results with proposal policies across three environments, averaged over three random seeds.

the TDM model exhibits negative transfer in the MPC with proposal policies setting, despite showing positive transfer in transition prediction and off-policy evaluation.

In the random shooting setting, the planner's limited ability to sample high-quality actions hinders its effective utilization of model predictions, leading to consistently poor performance across all world models. Nevertheless, TrajWorld demonstrates comparatively better results under these limitations. See Appendix C.4 for detailed results.

### 5.5. Analysis

**Few-shot adaptation.** TrajWorld presents pre-training benefits in few-shot scenarios. In Figure 8a, we show the prediction error across varying levels of data scarcity and compare TrajWorld with and without pre-training. These results highlight that the advantages of pre-training become increasingly pronounced as data becomes more limited.

**Discretization visualization.** We use t-SNE (van der Maaten & Hinton, 2008) to visualize the linear weights of our model's prediction head for each category. The mapped weights exhibit strong continuity in Figure 8b. Since the output categories' indices are aligned with the bins in increasing order, this indicates that our model has learned the ordering of bins shared by variants, despite being trained via an unordered classification objective. This suggests the model's potential for fine interpolation between existing bins and extrapolation to unseen ranges of variant values.

**Variate attention visualization.** We visualize the variate attention maps of our fine-tuned model in the Walker2D environment, whose states are ordered with joint positions first, followed by their velocities. As shown in Figure 8c, the attention map exhibits prominent diagonal patterns that focus on the corresponding joint's position and velocity, suggesting the model's understanding of each variate's semantics. Additionally, the strong attention between neighboring variate, such as physically linked joints, further confirms the model's grasp of joint relationships. We also observe notable attention patterns between states and actions, and these additional results are available in Appendix C.6.

**Effects of pre-training scale and diversity** We assess the impact of dataset scale and diversity by pre-training three TrajWorld variants on subsets of UniTraj: a 1/10 sample, a 1/100 sample, and the JAT subset (purely expert trajectories from five environments), followed by fine-tuning on downstream environments. For transition prediction, we adopt a challenging setup where models are fine-tuned on expert data per environment but evaluated across all data levels. Model-predictive control (MPC) results are also reported. As shown in Figure 9, all subset-pretrained models outperform training from scratch, but fall short of the model trained on the full UniTraj dataset. This reveals a desirable scaling trend: larger and more diverse pre-training data consistently lead to better generalization. These findings highlight the value of heterogeneous pre-training on large-scale datasets.

## 6. Related Work

**Trajectory dataset.** Data-driven approaches for control like imitation learning (Florence et al., 2022; Shafiullah et al., 2022; Gallouédec et al., 2024) and offline reinforcement learning (Fu et al., 2020; Rafailov et al., 2024; Gulcehre et al., 2020; Qin et al., 2022) have promoted the public availability of trajectory datasets. However, these datasets are rarely utilized as unified big data for foundation models, likely due to their isolated characteristics, such as differences in policy levels, observation spaces, and action spaces. In fact, the largest robotics dataset, Open X-Embodiment (O'Neill et al., 2024), is typically used for imitation learning with homogeneous visual observations and end-effector actions (Team et al., 2024; Kim et al., 2024). Gato (Reed et al., 2022) collects a large-scale dataset across diverse environments for a generalist agent, but it is not publicly available. In contrast, we curate public heterogeneous datasets, targeting a more capable trajectory world model.

**Cross-environment architecture.** Zero-padding to fit a maximum length (Yu et al., 2020a; Hansen et al., 2024; Schmied et al., 2024; Seo et al., 2022) or using separate neural network heads (Wang et al., 2024a; D'Eramo et al., 2020) hinders knowledge transfer between heterogeneous environ-

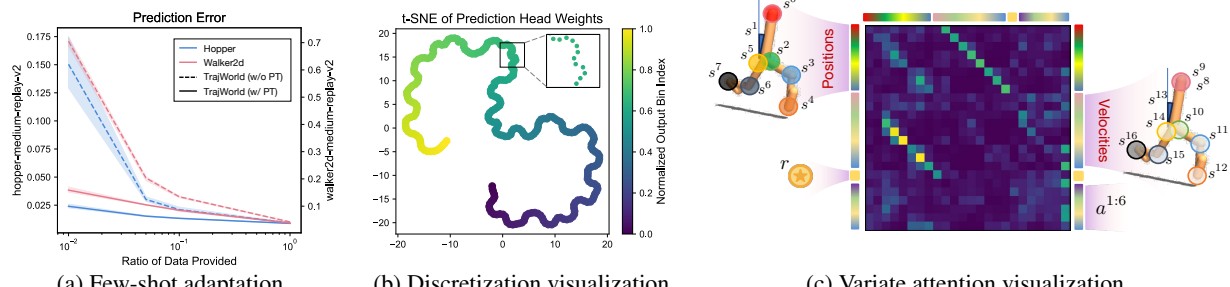

Figure 8. Model analysis. (a) Downstream prediction error of TrajWorld under varying data scarcity levels. (b) t-SNE visualization of the linear weights in the model's prediction head. (c) Variate attention map from the third layer of TrajWorld fine-tuned on Walker2D.

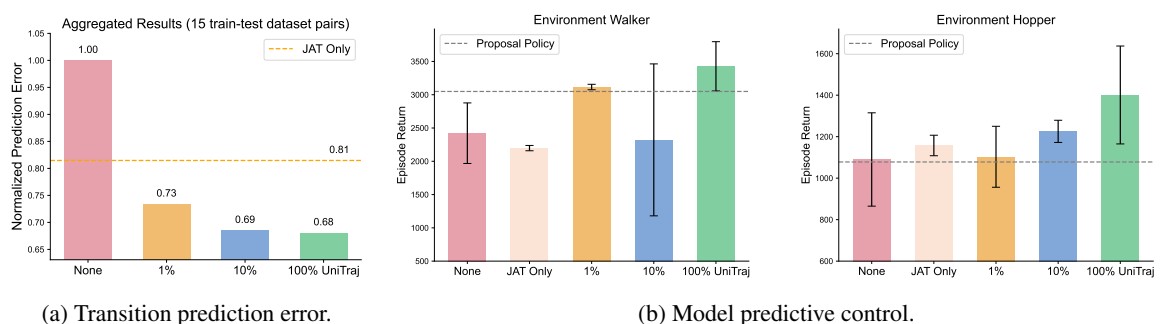

Figure 9. Effects of pre-training scale and diversity. (a) Aggregated transition prediction error. (b) Model predictive control performance on Walker2D and Hopper. All results are obtained from models fine-tuned from pre-trained TrajWorld on different subsets of UniTraj.

ments with mismatched or differently sized state and action spaces. Previous work has resorted to flexible architectures like graph neural networks (Huang et al., 2020; Kurin et al., 2021) and Transformers (Gupta et al., 2022; Hong et al., 2021) for policy learning. Our method leverages a similar architecture for world modeling (Janner et al., 2021; Zhang et al., 2021), but introduces the two-dimensional attention design for the first time in this context. More importantly, no for computational efficiency, as in prior work of other fields (Ho et al., 2019; Arnab et al., 2021; Nayakanti et al., 2023), we show its benefits for cross-environment transfer.

**World model pre-training.** The homogeneity of videos across diverse tasks, environments, and even embodiments has driven rapid advancements in large-scale video pre-training for world models (Seo et al., 2022; Wu et al., 2024a;b; Ye et al., 2024; Cheang et al., 2024). However, heterogeneity across different sets of sensors and actuators poses significant challenges to developing general world models based on low-dimensional sensor information.

Our work is particularly relevant to Schubert et al. (2023), which trains a generalist transformer dynamics model from 80 heterogeneous environments. Still, they only observe positive transfer when adapting to a simple cart-pole environment and fail for a more complex walker environment. In contrast, our work, for the first time, validates the positive transfer benefits across such more complex environments.

## 7. Conclusion

We address the challenge of building large-scale pre-trained world models for heterogeneous environments with distinct sensors, actuators, and dynamics. Our contributions include UniTraj, a dataset of over one million trajectories from 80 environments, and TrajWorld, a flexible architecture for cross-environment transfer. Pre-training TrajWorld on Uni-Traj achieves superior results in transition prediction and off-policy evaluation, demonstrating the first successful transfer of world models across complex control environments.

**Limitations and future work.** While this work takes a successful first step, there is significant room for further study. Despite the strong practical performance, one limitation of our architecture is that the discretization scheme constrains predictions to a fixed range, making it theoretically difficult to model extremely out-of-distribution transitions beyond these bounds. Additionally, our model, designed for scalable pre-training, has a larger capacity compared to classic MLPs, which poses challenges in model calibration (Guo et al., 2017), particularly in scenarios where uncertainty quantification is critical, such as offline RL (Yu et al., 2020b). This increased complexity also comes with additional computational costs. For future work, we envision that pre-training multimodal world models incorporating both visual and proprioceptive observations could lead to models with a deeper understanding of the physical world.

# Acknowledgements

This work was supported by the National Natural Science Foundation of China (U2342217 and 62021002), the BNRist Innovation Fund (BNR2024RC01010), and the National Engineering Research Center for Big Data Software.

# Impact Statement

This paper presents work whose goal is to advance the field of Machine Learning. There are many potential societal consequences of our work, none which we feel must be specifically highlighted here.

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

# A. UniTraj Dataset Details

## A.1. Overview of UniTraj Components

In this part, we provide a brief overview of each component of the UniTraj dataset.

**ExORL (Yarats et al., 2022).** Exploratory Data for Offline RL (ExORL) follows a two-step data collection protocol. First, data is generated in reward-free environments using unsupervised exploration strategies (Laskin et al., 2021). Next, this data is relabeled with either a standard or hand-designed reward function specific to each environment's task. This procedure leads to data with broader state-action space coverage, which benefits generalization-demanding scenarios like offline RL.

**RL Unplugged (Gulcehre et al., 2020).** We incorporate RL Unplugged's dataset from the DeepMind Control Suite domains. Most of the data collected in this domain are generated by recording D4PG's training runs (Barth-Maron et al., 2018), while Manipulator insert ball and Manipulator insert peg's data is collected using V-MPO (Song et al., 2020).

**JAT (Gallouédec et al., 2024).** We utilize Jack of All Trades (JAT)'s released dataset, which is collected using expert RL agent's rollouts. These agents are trained using asynchronous PPO (Schulman et al., 2017), following the Sample Factory implementation (Petrenko et al., 2020). Specifically, we only use the subset of the dataset that was collected in the OpenAI Gym environments, excluding data collected in Walker2D, HalfCheetah, and Hopper.

**DB-1 (Wen et al., 2022).** The dataset for Digital Brain-1 (DB-1), a reproduction of Gato (Reed et al., 2022), also consists solely of expert policy rollouts. Although the released dataset contains only five expert episodes per domain, it spans multiple environments, including various DeepMind Control Suite environments and custom ones from Modular RL.

**TD-MPC2 (Hansen et al., 2024).** TD-MPC2 is a state-of-the-art model-based RL algorithm. We include released data from single-task TD-MPC2 agents' replay buffers, collected from DeepMind Control Suite environments.

**Modular RL (Huang et al., 2020).** The Modular RL environments introduced by Huang et al. (2020) feature customizable embodiments with varying limb and joint configurations. We collected the data on these environments by ourselves. Specifically, we used the provided XML files to define different embodiment structures and followed the original reward function designs. We ran the TD3 algorithm (Fujimoto et al., 2018) and stored all episodes until the policy began to converge. The hyperparameters for TD3 are kept consistent with the default settings provided in the official repository repository[2].

## A.2. List of Environments

The curated UniTraj dataset spans a diverse range of environments from multiple sources, including DeepMind Control Suite, OpenAI Gym, and various customized environments. In Table 2, we provide a detailed list of environments used in each component of UniTraj.

## A.3. Sampling Weights

We manually weighted different subsets, trying to balance size and diversity. The sample weights are shown in Table 3.

# B. Experimental Details

## B.1. Model Implementation

**TrajWorld.** For discretization, as described in Section 4.2, we can employ two methods: one-hot encoding and Gaussian histograms. Specifically, the Gaussian histogram method is utilized for input discretization, while the one-hot encoding is applied for target discretization. Compared to one-hot encoding, Gaussian histograms provide a more fine-grained representation of value information. While we can also use Gaussian histograms for target discretization, one-hot encoding is more suitable for uncertainty quantization in future applications such as offline RL. This is because two Gaussian distributions with the same standard derivation can yield different entropy when discretized into histograms.

---

[2]https://github.com/sfujim/TD3

| Component | Environments |
|---|---|
| ExORL | Cartpole, Jaco, Quadruped, Walker[†] |
| RL Unplugged | Cartpole, Fish, Humanoid, Manipulator, Walker[†] |
| JAT | Double Pendulum[*], Pendulum[*], Pusher[*], Reacher[*], Swimmer[*] |
| DB-1 | Acrobat, Ball In Cup, Cartpole, Cheetah-2-back, Cheetah-2-front, Cheetah-3-back, Cheetah-3-balanced, Cheetah-3-front, Cheetah-4-allback, Cheetah-4-allfront, Cheetah-4-back, Cheetah-4-front, Cheetah-5-back, Cheetah-5-balanced, Cheetah-5-front, Cheetah-6-back, Cheetah-6-front, Finger, Fish, Hopper[†], Hopper-3, Hopper-5, Humanoid, Humanoid-2d-7-left-arm, Humanoid-2d-7-left-leg, Humanoid-2d-7-lower-arms, Humanoid-2d-7-right-arm, Humanoid-2d-7-right-leg, Humanoid-2d-8-left-knee, Humanoid-2d-8-right-knee, Humanoid-2d-9-full, Manipulator, Reacher, Swimmer6, Swimmer15, Walker[†], Walker-2-flipped, Walker-2-main, Walker-3-flipped, Walker-3-main, Walker-4-flipped, Walker-4-main, Walker-5-flipped, Walker-5-main, Walker-6-flipped, Walker-6-main |
| TD-MPC2 | Acrobot, Ball In Cup, Cartpole, Cheetah[†], Finger, Fish, Hopper[†], Pendulum[†], Reacher[†], Walker[†] |
| Modular RL | Cheetah-2-back, Cheetah-2-front, Cheetah-3-back, Cheetah-3-balanced, Cheetah-4-allback, Cheetah-4-back, Cheetah-4-front, Cheetah-5-back, Cheetah-5-balanced, Cheetah-5-front, Cheetah-6-back, Cheetah-6-front, Hopper-3, Hopper-5, Walker-2-flipped, Walker-3-flipped, Walker-4-flipped, Walker-5-flipped, Walker-6-flipped, Walker-7-flipped |

*Table 2.* A detailed list of environments used in the UniTraj dataset. For environments sharing the same name, we mark those from OpenAI Gym with an asterisk ($*$) and those from DeepMind Control Suite with a dagger ($†$). Notably, the Gym Hopper, Walker2D, and HalfCheetah environments used for evaluating our methods and baselines differ from their DeepMind Control Suite counterparts, exhibiting variations in state/action definitions and environment parameters.

| Subsets | ExORL | RLU | JAT | DB-1 | TD-MPC2 | Modular RL |
|---|---|---|---|---|---|---|
| (Unnormalized) sampling weight | 75 | 5 | 90 | 1 | 90 | 30 |

*Table 3.* Sampling weights of subsets for pre-training with UniTraj dataset.

For prediction, each bin $[b_{i-1}, b_i]$ is represented by its center $c_i = (b_{i-1} + b_i)/2$. Given the predicted bin probability $p_i$, the output value distribution can be expressed as $P(X = x) = \sum_{i=1}^{B} p_i \mathbf{1}(x = c_i)$ or $P(X = x) = \sum_{i=1}^{B} p_i \mathbf{1}(b_{i-1} < x \leq b_i)/(b_i - b_{i-1})$. We use the former for simplicity.

When pre-training with data from heterogeneous environments, for practical reasons, each batch is made up of data from a single environment.

We provide the hyperparameters used in pre-training and fine-tuning in Table 4. On transition prediction and OPE experiments, the environment-specific models trained from scratch use the same set of hyperparameters as fine-tuning.

**Baseline: Transformer Dynamics Model (TDM).** TDM (Schubert et al., 2023) does not provide an official implementation. To enable a fair comparison, we adapt our TrajWorld implementation to reproduce TDM while maintaining consistency in discretization and embedding methods. Furthermore, when trained using a cross-entropy loss, we mask actions and require the model to only predict the next states and rewards—unlike the TDM paper, where all variates are predicted. During inference, the model predicts each scalar dimension of the state sequentially, followed by setting each scalar of the action (e.g., provided by the policy in off-policy evaluation) one at a time. The hyperparameters for pre-training and fine-tuning are kept consistent with those used in TrajWorld (Table 4), except for the batch size for pre-training. Due to GPU memory constraints, the batch size for pre-training, originally set to 64, is reduced to 16. Like TrajWorld, we use the same hyperparameters as fine-tuning for environment-specific models trained from scratch.

| | Hyperparameter | Value |
|---|---|---|
| Architecture | Input discretization | Gauss-hist |
| | Target discretization | One-hot |
| | Transformer blocks number | 6 |
| | Attention heads number | 4 |
| | Transformer context length | 20 |
| | Hidden dimension | 256 |
| | MLP hidden | [1024,256] |
| | MLP activation | GeLU |
| Pre-training | Total gradient steps | 1M |
| | Batch size | 64 |
| | Learning rate | $1 \times 10^{-4}$ |
| | Dropout rate | 0.05 |
| | Optimizer | Adam |
| | Weight decay | $1 \times 10^{-5}$ |
| | Gradient clip norm | 0.25 |
| | Scheduler | Warmup cosine decay |
| | Scheduler warmup steps | 10000 |
| Fine-tuning | Total max gradient steps | 1.5M |
| | Max epochs | 300 |
| | Steps per epoch | 5000 |
| | Batch size | 64 |
| | Learning rate | $1 \times 10^{-5}$ |
| | Dropout rate | 0.05 |
| | Optimizer | Adam |
| | Weight decay | $1 \times 10^{-5}$ |
| | Gradient clip norm | 0.25 |
| | Scheduler | Warmup cosine decay |
| | Scheduler warmup steps | 10000 |

*Table 4.* Hyperparameters for TrajWorld.

**Baseline: MLP Ensemble.** Following prior work (Chua et al., 2018; Janner et al., 2019; Yu et al., 2020b), we train an ensemble of transition models, parameterized as a diagonal Gaussian distribution of the next state and reward, implemented using MLPs. These models are trained with bootstrapped training samples, and optimized via negative log-likelihood. After training, we select an elite subset of models based on validation loss, and during inference, a model from this subset is randomly sampled for predictions. For pre-training on heterogeneous environments, we implement the MLP Ensemble baseline by padding each state vector to 90 dimensions and each action vector to 30 dimensions, resulting in a 120-dimensional input to the MLP. The model outputs the distribution over a 91-dimensional vector (90 for the next state and 1 for the reward). To ensure a fair comparison with other methods, we match the parameter count of the ensemble to TrajWorld, and no environment identities are provided to this baseline. The hyperparameters are listed in Table 5. Environment-specific models trained from scratch use the same hyperparameters as in fine-tuning.

### B.2. Zero-Shot Generalization

#### B.2.1. ENVIRONMENT PARAMETER TRANSFER

**Pendulum.** We pre-train the TrajWorld model on 60 Gym Pendulum environments, where the gravity values range from 8 m/s$^2$ to 12 m/s$^2$. The pre-training dataset is collected by running the TD3 algorithm (Fujimoto et al., 2018) and storing all episodes until the policy converges. For evaluation, we use five holdout environments with gravity values between 6.5m/s$^2$ and 7.5m/s$^2$, collecting data in the same manner as the training datasets. The zero-shot results are reported as the average prediction error on these holdout datasets.

| | Hyperparameter | Value |
|---|---|---|
| Architecture | MLP hidden | [640, 640, 640, 640] |
| | Ensemble number | 7 |
| | Ensemble elite Number | 5 |
| Pre-training | Total gradient steps | 1M |
| | Batch size | 256 |
| | Learning rate | $1 \times 10^{-4}$ |
| | Optimizer | Adam |
| Fine-tuning | Total max gradient steps | 1.5M |
| | Max epochs | 300 |
| | Steps per epoch | 5000 |
| | Batch size | 256 |
| | Learning rate | $1 \times 10^{-5}$ |
| | Optimizer | Adam |

*Table 5.* Hyperparameters for MLP Ensemble.

**Walker2D.** We pre-train a four-layer TrajWorld model using 45 training datasets provided by MACAW (Mitchell et al., 2021) and evaluate it on a separate dataset also from MACAW. The datasets in MACAW are collected under varying physical conditions, including differences in body mass, friction, damping, and inertia.

### B.2.2. CROSS-ENVIRONMENT TRANSFER

We evaluate the model pre-trained on UniTraj by performing a ten-step rollout in the Cart-2-Pole and Cart-3-Pole environment from the DeepMind Control Suite. The rollout is conditioned on a history of ten prior timesteps. After this initial context, actions are applied in a simple predefined manner: either continuously pushing to the right ($a = 0.5$) or to the left ($a = -0.5$). The action repeat for the Cart-2-Pole and Cart-3-Pole environment is set to 4.

### B.3. Transition Prediction

The model is trained on a dataset using this dataset's training set and tested on five test datasets that come from the same environment. The evaluation for each test set is based on the model's prediction error across the entire test dataset. We use the Mean Absolute Error (MAE) as the evaluation metric. The prediction of TrajWorld is done by maintaining a history context window of 19 to predict the 20th state and reward.

In Figure 1, the prediction errors for each train-test dataset pair are normalized by dividing them by the MAE of the TrajWorld model without pre-training. The final result is then obtained by averaging across all environments.

### B.4. Off-Policy Evaluation

### B.4.1. IMPLEMENTATION: MODEL-BASED OPE

Given a world model, the most direct method for off-policy evaluation (OPE) is Monte Carlo policy evaluation. This involves starting from a set of initial states, performing policy rollouts within the learned model, and averaging the accumulated rewards to estimate the policy value. The procedure is summarized in Algorithm 1.

In practice, we use a discount factor of $\gamma = 0.995$ and a horizon length of $h = 2000$. The number of samples $N$ is set such that each trajectory's initial state from the behavior dataset is used exactly once, resulting in approximately $N \approx 1000$. We use KV cache to accelerate the rollouts of our TrajWorld.

### B.4.2. BASELINES

We primarily compare against model-based OPE with **Energy-based Transition Models (ETM)** (Chen et al., 2024), a strong baseline that significantly outperforms previous methods and represents state-of-the-art on the DOPE benchmark (Fu et al., 2021).

---

**Algorithm 1** Model-Based OPE

---

**Input:** learned world model $P_\theta(s_{t+1}, r_{t+1}|s_t, a_t)$, test policy $\pi$, samples number $N$, initial state distribution $S_0$, discount factor $\gamma$, horizon length $h$.
**for** $i = 1$ to $N$ **do**
$\quad R_i \leftarrow 0$
$\quad$ Sample initial state $s_0 \sim S_0$
$\quad$ **for** $t = 0$ to $h - 1$ **do**
$\quad\quad a_t \sim \pi(\cdot|s_t)$
$\quad\quad s_{t+1}, r_{t+1} \sim P_\theta(\cdot|s_t, a_t)$
$\quad\quad R_i \leftarrow R_i + \gamma^t r_{t+1}$
$\quad$ **end for**
**end for**
**Return** $\hat{V}(\pi) = \frac{1}{N}\sum_{i=1}^{N} R_i$

---

We also include five classic OPE methods as baselines from the DOPE benchmark: **Fitted Q-Evaluation (FQE)** (Le et al., 2019), **Doubly Robust (DR)** (Jiang & Li, 2016), **Importance Sampling (IS)**, (Kostrikov & Nachum, 2020) **DICE** (Yang et al., 2020), and **Variational Power Method (VPM)** (Wen et al., 2020).

### B.4.3. METRICS

We adopt the evaluation metrics used in the DOPE benchmark.

**Mean Absolute Error.** The absolute error quantifies the deviation between the true value and the estimated value of a policy, defined as:

$$\text{AbsErr} = |V^\pi - \hat{V}^\pi|, \tag{7}$$

where $V^\pi$ represents the true value of the policy, and $\hat{V}^\pi$ denotes its estimated value. The Mean Absolute Error (MAE) is computed as the average absolute error across all evaluated policies. To aggregate results, these values are normalized by the difference between the maximum and minimum true policy values.

**Rank correlation.** Rank correlation, also known as Spearman's rank correlation coefficient ($\rho$), measures the ordinal correlation between the estimated policy values and their true values. It is given by:

$$\text{RankCorr} = \frac{\text{Cov}(V_{1:N}^\pi, \hat{V}_{1:N}^\pi)}{\sigma(V_{1:N}^\pi)\sigma(\hat{V}_{1:N}^\pi)}, \tag{8}$$

where $1:N$ represents the indices of the evaluated policies.

**Regret@$k$.** Regret@$k$ quantifies the performance gap between the actual best policy and the best policy selected from the top-$k$ candidates (ranked by estimated values). It is formally defined as:

$$\text{Regret@}k = \max_{i \in 1:N} V_i^\pi - \max_{j \in \text{topk}(1:N)} V_j^\pi \tag{9}$$

where $\text{topk}(1:N)$ denotes the indices of the top $k$ policies based on estimated values $\hat{V}^\pi$. In our experiments, we specifically use **normalized Regret@1** as the evaluation metric.

### B.5. Model Predictive Control

We evaluate model predictive control (MPC) performance under two planning settings: policy proposal and random shooting.

**Policy proposal setting.** Action candidates are generated by perturbing the output of a learned action policy with Gaussian noise. Specifically, we first query the policy to obtain a mean action sequence and then add zero-mean Gaussian noise to each action in the sequence. This results in a set of diverse trajectories centered around the policy's behavior.

**Random shooting setting.** Candidate action sequences are sampled directly from a Gaussian distribution without guidance from a learned policy. Each trajectory is independently sampled by drawing actions from a zero-mean Gaussian distribution with a fixed standard deviation.

**Hyperparameters.** For both settings, we use a sample size of 128 candidate trajectories per MPC rollout, across all environments. The best-performing action sequence is selected based on predicted cumulative reward computed using the world model.

The planning horizon is set based on the characteristics of each environment. Specifically, we use a horizon of 25 steps for both HalfCheetah and Walker2D, while a longer horizon of 50 steps is adopted for Hopper. This extended horizon for Hopper helps mitigate short-sighted planning behavior, which is particularly detrimental in this more fragile environment.

To ensure optimal performance across different world models, the standard deviation of the Gaussian noise used for action sampling is tuned individually for each environment. The noise level is set to 0.05 for Hopper, 0.2 for Walker2D, and 0.025 for HalfCheetah. These values were empirically selected to balance exploration and stability during trajectory sampling.

These settings are used consistently in all experiments involving MPC in this work. The same configurations are applied for evaluating all world models, ensuring fair comparison.

### B.6. Computational Cost

Our implementation, built upon JAX (Bradbury et al., 2018), benefits from significant computational efficiency. Both pre-training and fine-tuning of the TrajWorld model can be conducted on a single 24GB NVIDIA RTX 4090 GPU. For comparison, the computational cost for 1.5M training steps in our implementations of the MLP Ensemble, TDM, and TrajWorld is 1.5, 36, and 28 hours, respectively. This highlight that TrajWorld achieves strong performance with lower computational cost than TDM.

## C. Extended Experimental Results

### C.1. Detailed Prediction Error for Baselines

We report the prediction error for MLP Ensemble and TDM in Figure 10 and 11, respectively.

### C.2. Quantitative Results for Off-Policy Evaluation

We report the raw absolute error, rank correlation and regret@1 for each OPE method and task in Table 6.

### C.3. Off-Policy Evaluation with Pre-trained Models on Parameter-Variant Environments.

In addition to the zero-shot prediction error reported in Section 5.1, we further investigate our four-layer model pre-trained on Walker2D with variant friction, mass, etc. Specifically, we evaluate the model's performance by fine-tuning it and testing it on downstream off-policy evaluation tasks on standard Walker2D. The results are summarized in Table 7. This provides additional evidence, beyond the zero-shot prediction error, demonstrating that TrajWorld exhibits strong capability for transfer to environments with varying parameters.

| Env. | Level | TrajWorld (w/o PT) | TrajWorld (w/ PT) |
|---|---|---|---|
| | random | 262 ± 34 | **76 ± 6** |
| | medium | 68 ± 2 | **40 ± 4** |
| Walker2D | m-replay | 71 ± 11 | **46 ± 1** |
| | m-expert | **49 ± 1** | 76 ± 1 |
| | expert | 281 ± 8 | **186 ± 1** |

*Table 7.* Raw absolute error of off-policy evaluation for a four-layer TrajWorld model trained from scratch compared to a model fine-tuned from a pre-trained version on the Walker2D dataset with variant environment parameters with holdout onest, averaged over two seeds.

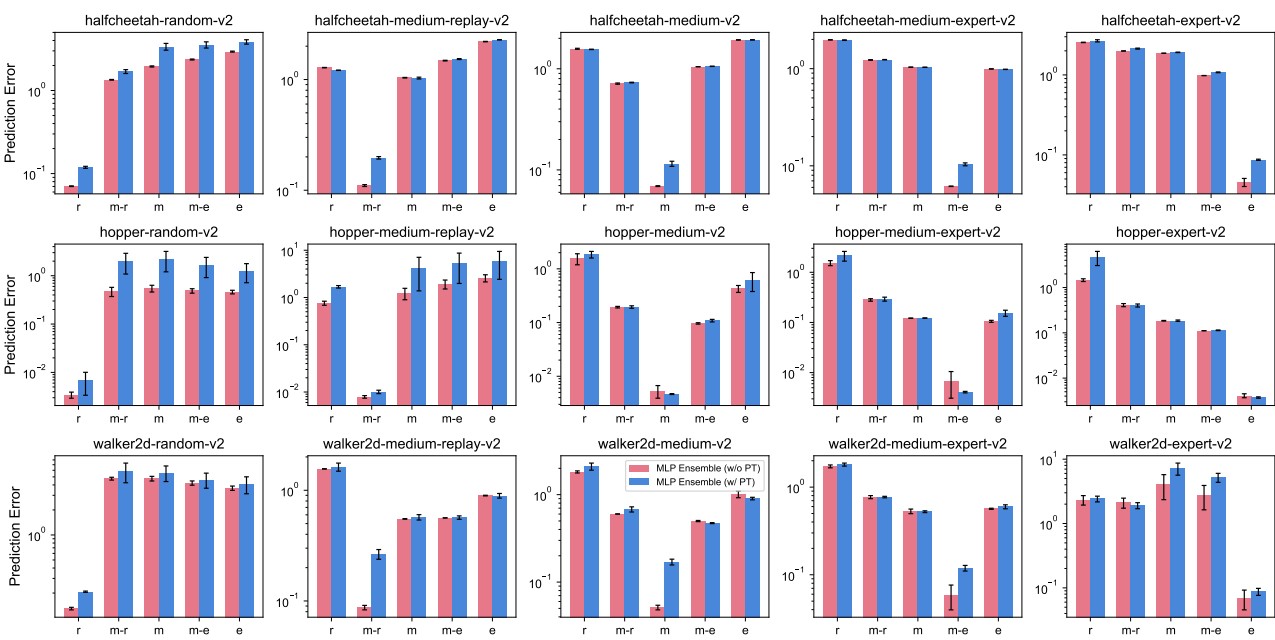

*Figure 10.* Mean absolute errors (MAE) of transition prediction for MLP Ensemble, with and without pre-training (PT), across different train-test dataset pairs. Each subplot corresponds to a distinct training dataset, with the test datasets shown on the x-axis (r=random, m-r=medium-replay, m=medium, m-e=medium-expert, e=expert). Error bars represent the standard deviation across three random seeds.

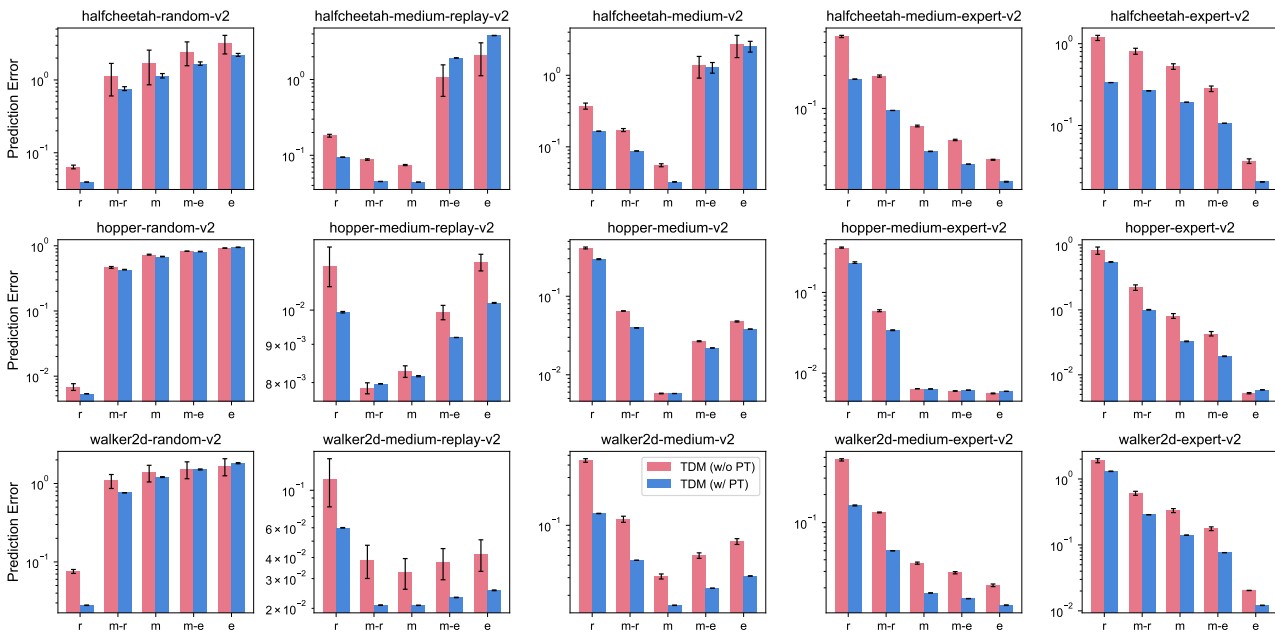

*Figure 11.* Mean absolute errors (MAE) of transition prediction for TDM, with and without pre-training (PT), across different train-test dataset pairs. Each subplot corresponds to a distinct training dataset, with the test datasets shown on the x-axis (r=random, m-r=medium-replay, m=medium, m-e=medium-expert, e=expert). Error bars represent the standard deviation across three random seeds.

| Env. | Level | ETM | MLP (w/o PT) | MLP (w/ PT) | TDM (w/o PT) | TDM (w/ PT) | TW (w/o PT) | TW (w/ PT) |
|---|---|---|---|---|---|---|---|---|
| **Hopper** | random | 236 ± 15 | 245 ± 9 | 307 ± 15 | 79 ± 19 | 160 ± 17 | 259 ± 27 | **98 ± 1** |
| | medium | **47 ± 21** | 149 ± 30 | 181 ± 18 | 140 ± 10 | 145 ± 7 | 81 ± 11 | 127 ± 9 |
| | m-replay | **29 ± 8** | **24 ± 5** | 33 ± 2 | 38 ± 13 | 56 ± 13 | 60 ± 7 | 73 ± 6 |
| | m-expert | 32 ± 4 | 87 ± 35 | 173 ± 15 | 116 ± 21 | 79 ± 3 | **48 ± 7** | 69 ± 8 |
| | expert | 71 ± 16 | 167 ± 36 | 218 ± 29 | 283 ± 8 | 100 ± 2 | 105 ± 31 | **42 ± 2** |
| **Walker2D** | random | 339 ± 10 | 356 ± 4 | 372 ± 3 | 291 ± 40 | **264 ± 9** | 312 ± 19 | **269 ± 1** |
| | medium | 159 ± 13 | 181 ± 10 | 371 ± 9 | 104 ± 22 | 123 ± 12 | **61 ± 6** | 101 ± 7 |
| | m-replay | 132 ± 31 | 131 ± 8 | 313 ± 15 | 143 ± 52 | 147 ± 3 | **54 ± 12** | 182 ± 10 |
| | m-expert | 152 ± 9 | 210 ± 47 | 340 ± 19 | 87 ± 24 | 137 ± 17 | **60 ± 11** | **72 ± 7** |
| | expert | 364 ± 7 | 344 ± 20 | 368 ± 15 | 403 ± 141 | 458 ± 19 | 272 ± 124 | **100 ± 2** |
| **Halfcheetah** | random | **842 ± 42** | 965 ± 2 | 1137 ± 27 | 1079 ± 11 | 1050 ± 4 | 1028 ± 17 | 1059 ± 7 |
| | medium | 655 ± 114 | 734 ± 24 | 973 ± 91 | 1435 ± 54 | 1312 ± 21 | 568 ± 23 | **444 ± 4** |
| | m-replay | 727 ± 119 | 712 ± 59 | 993 ± 41 | 927 ± 261 | 730 ± 25 | **540 ± 45** | **540 ± 16** |
| | m-expert | 689 ± 203 | 692 ± 65 | 1117 ± 90 | 923 ± 98 | 1319 ± 23 | 809 ± 150 | **528 ± 10** |
| | expert | 758 ± 116 | 973 ± 175 | 1243 ± 36 | 1273 ± 158 | **646 ± 50** | 1013 ± 246 | 841 ± 14 |

(a) Raw absolute error

| Env. | Level | ETM | MLP (w/o PT) | MLP (w/ PT) | TDM (w/o PT) | TDM (w/ PT) | TW (w/o PT) | TW (w/ PT) |
|---|---|---|---|---|---|---|---|---|
| **Hopper** | random random | 0.61 ± 0.15 | 0.65 ± 0.17 | 0.43 ± 0.09 | **0.90 ± 0.05** | 0.82 ± 0.05 | 0.56 ± 0.23 | 0.81 ± 0.01 |
| | medium | **0.94 ± 0.04** | 0.81 ± 0.05 | 0.72 ± 0.03 | 0.64 ± 0.10 | 0.46 ± 0.07 | 0.81 ± 0.06 | 0.31 ± 0.10 |
| | m-replay | **0.97 ± 0.02** | 0.99 ± 0.00 | **0.98 ± 0.00** | 0.96 ± 0.01 | 0.89 ± 0.04 | 0.86 ± 0.05 | 0.61 ± 0.36 |
| | m-expert | **0.95 ± 0.01** | **0.90 ± 0.09** | 0.79 ± 0.05 | 0.55 ± 0.32 | 0.86 ± 0.01 | 0.93 ± 0.01 | 0.87 ± 0.02 |
| | expert | 0.85 ± 0.05 | 0.62 ± 0.07 | 0.42 ± 0.09 | -0.34 ± 0.17 | 0.78 ± 0.04 | 0.86 ± 0.04 | **0.95 ± 0.00** |
| **Walker2D** | random | -0.12 ± 0.32 | 0.75 ± 0.03 | 0.58 ± 0.17 | 0.73 ± 0.10 | 0.79 ± 0.02 | 0.67 ± 0.01 | **0.78 ± 0.00** |
| | medium | 0.78 ± 0.12 | 0.90 ± 0.03 | 0.44 ± 0.12 | 0.86 ± 0.04 | 0.91 ± 0.02 | **0.95 ± 0.01** | **0.94 ± 0.00** |
| | m-replay | 0.77 ± 0.10 | **0.95 ± 0.01** | 0.72 ± 0.08 | 0.88 ± 0.03 | 0.93 ± 0.02 | **0.97 ± 0.02** | 0.77 ± 0.01 |
| | m-expert | 0.67 ± 0.14 | 0.92 ± 0.02 | 0.74 ± 0.06 | 0.91 ± 0.04 | 0.79 ± 0.06 | **0.95 ± 0.01** | **0.96 ± 0.01** |
| | expert | 0.54 ± 0.11 | 0.36 ± 0.11 | 0.11 ± 0.42 | 0.36 ± 0.42 | 0.80 ± 0.01 | 0.59 ± 0.30 | **0.94 ± 0.01** |
| **Halfcheetah** | random | 0.76 ± 0.10 | 0.90 ± 0.01 | **0.84 ± 0.12** | 0.93 ± 0.00 | 0.90 ± 0.00 | 0.91 ± 0.00 | **0.94 ± 0.00** |
| | medium | 0.78 ± 0.12 | 0.93 ± 0.01 | 0.93 ± 0.01 | -0.29 ± 0.38 | 0.14 ± 0.08 | 0.96 ± 0.00 | **0.98 ± 0.00** |
| | m-replay | 0.77 ± 0.10 | 0.90 ± 0.00 | 0.88 ± 0.02 | **0.93 ± 0.03** | 0.86 ± 0.02 | **0.93 ± 0.00** | 0.93 ± 0.01 |
| | m-expert | 0.91 ± 0.03 | **0.96 ± 0.02** | 0.90 ± 0.00 | 0.75 ± 0.10 | 0.27 ± 0.07 | 0.91 ± 0.06 | **0.97 ± 0.00** |
| | expert | 0.81 ± 0.10 | **0.90 ± 0.06** | 0.24 ± 0.44 | 0.24 ± 0.20 | 0.81 ± 0.02 | 0.49 ± 0.26 | **0.94 ± 0.01** |

(b) Rank correlation

| Env. | Level | ETM | MLP Ens.(w/o PT) | MLP Ens.(w/ PT) | TDM (w/o PT) | TDM (w/ PT) | TW (w/o PT) | TW (w/ PT) |
|---|---|---|---|---|---|---|---|---|
| **Hopper** | random | **0.20 ± 0.10** | **0.30 ± 0.28** | 0.62 ± 0.00 | **0.20 ± 0.15** | **0.25 ± 0.08** | **0.39 ± 0.32** | 0.37 ± 0.00 |
| | medium | **0.05 ± 0.04** | **0.03 ± 0.04** | **0.05 ± 0.04** | **0.17 ± 0.17** | 0.16 ± 0.00 | 0.22 ± 0.13 | **0.11 ± 0.06** |
| | m-replay | **0.00 ± 0.00** | 0.08 ± 0.00 | 0.08 ± 0.00 | 0.09 ± 0.02 | 0.08 ± 0.00 | 0.15 ± 0.01 | 0.14 ± 0.00 |
| | m-expert | **0.08 ± 0.07** | **0.09 ± 0.02** | **0.12 ± 0.21** | 0.37 ± 0.00 | **0.08 ± 0.00** | 0.15 ± 0.13 | **0.08 ± 0.00** |
| | expert | **0.08 ± 0.02** | **0.17 ± 0.17** | **0.17 ± 0.17** | 0.68 ± 0.55 | **0.08 ± 0.00** | 0.12 ± 0.15 | **0.10 ± 0.02** |
| **Walker2D** | random | 0.16 ± 0.09 | **0.05 ± 0.01** | 0.41 ± 0.40 | 0.10 ± 0.04 | **0.05 ± 0.00** | 0.17 ± 0.00 | 0.08 ± 0.00 |
| | medium | **0.00 ± 0.00** | 0.08 ± 0.00 | 0.28 ± 0.00 | 0.08 ± 0.07 | 0.12 ± 0.00 | **0.04 ± 0.07** | 0.08 ± 0.00 |
| | m-replay | **0.00 ± 0.00** | 0.07 ± 0.02 | 0.19 ± 0.16 | 0.08 ± 0.04 | 0.10 ± 0.06 | **0.03 ± 0.05** | **0.00 ± 0.00** |
| | m-expert | **0.03 ± 0.02** | 0.08 ± 0.00 | **0.09 ± 0.16** | **0.06 ± 0.10** | 0.12 ± 0.00 | **0.05 ± 0.05** | 0.08 ± 0.00 |
| | expert | **0.05 ± 0.05** | **0.19 ± 0.16** | **0.19 ± 0.26** | **0.12 ± 0.14** | 0.28 ± 0.00 | **0.28 ± 0.28** | **0.17 ± 0.10** |
| **Halfcheetah** | random | 0.20 ± 0.10 | 0.15 ± 0.10 | 0.11 ± 0.12 | **0.04 ± 0.00** | **0.09 ± 0.08** | **0.14 ± 0.10** | **0.03 ± 0.02** |
| | medium | **0.08 ± 0.08** | 0.18 ± 0.00 | 0.17 ± 0.02 | 0.70 ± 0.52 | 0.23 ± 0.07 | **0.12 ± 0.11** | **0.12 ± 0.02** |
| | m-replay | **0.16 ± 0.12** | **0.15 ± 0.10** | **0.23 ± 0.07** | **0.16 ± 0.05** | 0.25 ± 0.00 | **0.18 ± 0.00** | **0.18 ± 0.00** |
| | m-expert | 0.11 ± 0.10 | **0.06 ± 0.11** | 0.18 ± 0.00 | 0.16 ± 0.05 | 0.37 ± 0.00 | 0.14 ± 0.13 | **0.00 ± 0.00** |
| | expert | 0.12 ± 0.07 | 0.16 ± 0.02 | **0.04 ± 0.00** | 0.27 ± 0.10 | 0.14 ± 0.00 | 0.18 ± 0.03 | **0.20 ± 0.19** |

(c) Regret@1

*Table 6.* Quantitative results of all model-based methods (TW=TrajWorld) for OPE, averaged over 3 seeds.

| Env. | MLP (w/o PT) | MLP (w/ PT) | TDM (w/o PT) | TDM (w/ PT) | TW (w/o PT) | TW (w/ PT) | Proposal |
|---|---|---|---|---|---|---|---|
| **Hopper** | 948±61 | 1091±125 | 1287±26 | 1117±145 | 1090±225 | **1401±236** | *1078±143* |
| **Walker** | 3353±83 | **3465±20** | 3056±236 | 2619±36 | 2422±455 | **3427±370** | *3049±104* |
| **HalfCheetah** | 5645±10 | 5692±19 | 5611±85 | 5647±25 | **5858±17** | 5809±15 | *5697±30* |

*Table 8.* Quantitative results of all model-based methods (TW=TrajWorld) for MPC with action proposal, averaged over 3 seeds.

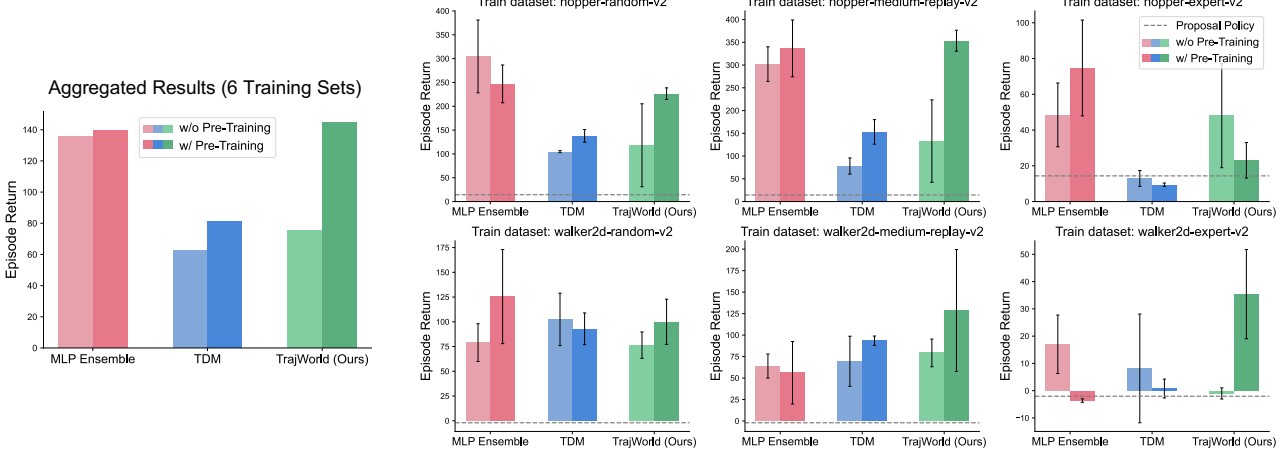

*Figure 12.* Model predictive control (MPC) results using a random shooting planner, averaged across three random seeds. The proposal policy line indicates the performance of a random action-sampling strategy.

## C.4. Additional Model Predictive Control Results

**Quantitative results with proposal policies.** We report quantitative results on MPC with action proposal in Table 8.

**MPC with random shooting planner.** Figure 12 presents MPC results using a random shooting planner with models trained on different datasets.

**Computational efficiecny.** TrajWorld predicts all variates jointly, unlike TDM which processes them sequentially. This leads to a major speedup: MPC for 1000 environment steps in HalfCheetah takes 40 minutes with TDM, but only 3 minutes with TrajWorld.

## C.5. Additional Zero-shot Cross-Environment Transfer

**Comparison with baselines.** We also provide zero-shot prediction from other baselines in Figure 14. As shown, in an unseen environment, both TDM and MLP baselines fail to generalize, producing incorrect predictions and failing to capture the underlying state-action relationship at all. Specifically, TDM fails to predict how push forces from two opposite directions lead to different x positions. On the other hand, MLP fails to produce any reasonable results with extreme error accumulation.

**Cart-3-pole environment.** We also test TrajWorld's zero-shot prediction on the more challenging Cart-3-pole environment, which has an 11-dimensional state space. Surprisingly, TrajWorld can still give

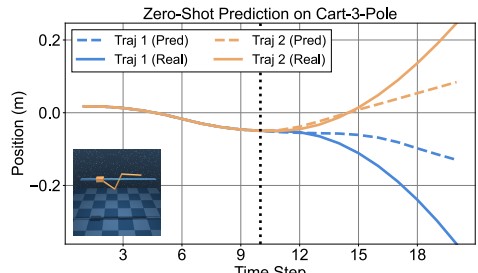

*Figure 13.* TrajWorld's zero-shot predictions for two Cart-3-Pole trajectories, which share 10 context steps but diverge due to differing subsequent actions.

cart's position predictions roughly aligned with the ground truth, despite not seeing this embodiment before. The action sequence is depicted in Section B.2.2.

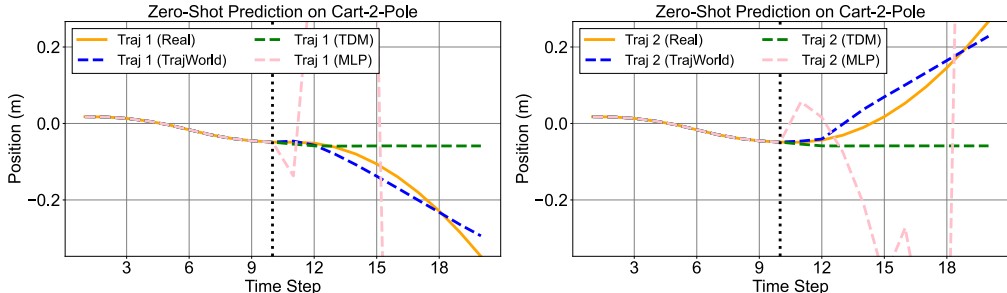

*Figure 14.* Zero-shot predictions from different pre-trained models on two Cart-2-Pole trajectories that share the same 10 context steps but diverge thereafter due to different future actions.

## C.6. Additional Variate Attention Visualization

We present the variate attention maps of our TrajWorld model across all six layers, comparing a fine-tuned model and a model trained from scratch, in Figures 15 and 16.

For the fine-tuned model, in the early layers (Layer 0 and 1), attention is more scattered and less structured, likely capturing broad and low-level features. In contrast, later layers (Layer 4 and 5) exhibit more focused attention, suggesting the model is concentrating on specific relationships or entities. The prominent diagonal patterns and neighboring attentions discussed in Section 5.5 can also be clearly observed in Layer 2. Additionally, diagonal patterns linking joint velocities and actions appear in Layers 1 and 2. Such diagonal patterns are also observed in the attention maps of the model trained from scratch.

A notable difference between the attention maps of the fine-tuned model and the model trained from scratch is the earlier emergence of diagonal patterns in the layers of the model trained from scratch. Specifically, while the first two layers of the fine-tuned model exhibit more scattered and less interpretable attention, the scratch-trained model immediately begins capturing structured diagonal patterns, particularly between positions and velocities, as well as velocities and actions. This probably suggests that pre-training changes the model's behavior. The model without pre-training tends to focus on environment-specific patterns and more localized features for prediction. In contrast, the fine-tuned model seems to dedicate its first two layers to extracting more semantically meaningful and generalizable features, encouraging the model to perform inference through in-context learning from these environment-agnostic representations.

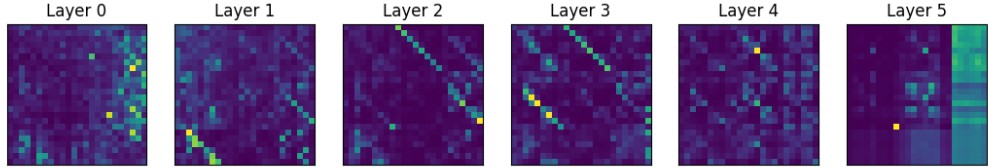

*Figure 15.* Variate attention maps of our pre-trained TrajWorld Model, fine-tuned under Walker2D environment.

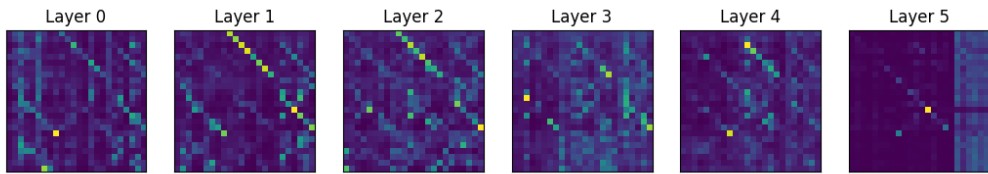

*Figure 16.* Variate attention maps of our TrajWorld Model in the Walker2D environment, trained from scratch.

## C.7. Additional Ablation Study on Pre-training Dataset

To investigate the contributions of different components of the UniTraj dataset to the pre-training process, we conduct an ablation study by training a four-layer TrajWorld model on a modified version of the UniTraj dataset, excluding two data

sources more closely aligned with the target environments: Modular RL and TD-MPC2. The results presented in Table 9 indicate that the advantages of pre-training stem from the diversity encompassed within the complete UniTraj dataset, rather than relying solely on data from domains closely resembling the target environments.

| Env. | Level | TrajWorld (w/o PT) | TrajWorld (w/ PT) |
|------|-------|--------------------|--------------------|
| Halfcheetah | medium | **491** ± 94 | **468** ± 40 |
| Walker2D | medium | 88 ± 21 | **54** ± 2 |
| Walker2D | expert | 141 ± 15 | **116** ± 12 |

(a) Raw absolute error

| Env. | Level | TrajWorld (w/o PT) | TrajWorld (w/ PT) |
|------|-------|--------------------|--------------------|
| Halfcheetah | medium | 0.95 ± 0.00 | **0.97** ± 0.01 |
| Walker2D | medium | 0.93 ± 0.01 | **0.97** ± 0.02 |
| Walker2D | expert | 0.56 ± 0.20 | **0.81** ± 0.04 |

(b) Rank Correlation

| Env. | Level | TrajWorld (w/o PT) | TrajWorld (w/ PT) |
|------|-------|--------------------|--------------------|
| Halfcheetah | medium | 0.18 ± 0.00 | **0.05** ± 0.07 |
| Walker2D | medium | 0.04 ± 0.06 | **0.00** ± 0.00 |
| Walker2D | expert | 0.34 ± 0.31 | **0.16** ± 0.16 |

(c) Regret@1

*Table 9.* OPE results for a four-layer TrajWorld model trained from scratch compared to a model fine-tuned from a pre-trained version on the ablation dataset, averaged over two seeds.

## D. Extended Discussion

**Limitations of bounded prediction.** Our discretization scheme (Section 4.2) has the drawback that it can only represent variate values within the bounded range $[b_0, b_B]$, restricted by the maximum and minimum in training data. This can lead to inaccurate predictions. For example, a model trained on trajectories from low-performing policies, may underestimate the reward of a high-rewarding transition. This may explain why our model slightly underperforms in Regret@1 for off-policy evaluation tasks. Since all variates share the same bin embeddings, a promising way to address this issue is to simply extend the value range of bins beyond the observed data limits for variates with narrow coverages. Although the model would not have encountered those out-of-range values for a specific variate during training, we hypothesize it could extrapolate similarly to regression models (e.g., MLPs), leveraging learned bin ordering shared with other variates. This hypothesis is supported by the bin continuity observed in Figure 8b. Further exploration and improvement of this approach are left for future work.

**Discussion with Schubert et al. (2023).** We demonstrate positive transfer to complex downstream environments such as Walker2D, not only for offline transition prediction and policy evaluation, but also for online MPC, which Schubert et al. (2023) did not. Our work differentiate from theirs in: (1) Setting: Instead of fine-tuning with $10^4$ episodes for MPC with random shooting, we more practically fine-tune with $10^2$ episodes for MPC with proposal policies; (2) Data diversity: Our UniTraj dataset emphasizes distribution diversity, rather than using pure expert trajectories; (3) Architecture: TrajWorld incorporates inductive biases tailored to the 2D structure of trajectory data for enhanced transferability. Notably, TDM exhibits negative transfer in our practical MPC setting. We believe our work complements and extends Schubert et al. (2023), offering new insights to the community.

