# OpenReview forum: "Trajectory World Models for Heterogeneous Environments"
_ICML.cc/2025/Conference — ICML 2025 poster_

### Official Review · Reviewer_bd1u · 2025-03-09

**Overall Recommendation:** 4

**Summary:**

This manuscript has 2 contributions:
1. A trajectory dataset UniTraj, a large-scale dataset including over one million trajectories collected from various distributions from 80 heterogeneous environments.
2. A Transformer-based architecture TrajWorld integrates interleaved variate and temporal attention mechanisms, aiming at transition prediction.
The authors first introduced the current challenge for trajectory prediction in a heterogeneous environment and their motivation. Then introduced the building process of dataset UniTraj, the architecture of TrajWorld model. The proposed model was tested on 15 datasets of 3 environments, which validated its performance.

**Claims And Evidence:**

Yes, I think this manuscript is well-organized and clearly stated.

**Essential References Not Discussed:**

n/a

**Experimental Designs Or Analyses:**

Yes. This proposed model used a two-way attention mechanism. The authors set a similar but with a one-dimensional attention model as the baseline, validating the good performance of the two-way attention mechanism

**Methods And Evaluation Criteria:**

Yes.

**Other Comments Or Suggestions:**

n/a

**Other Strengths And Weaknesses:**

I think the proposed dataset can be helpful for other researchers. And the proposed model can be a good baseline to study with.

**Questions For Authors:**

n/a

**Relation To Broader Scientific Literature:**

The large-scale dataset can be a helpful tool for broader scientific literature.

**Theoretical Claims:**

Yes. I think the authors clearly described the problem, related works, and their ideas.

---

> ### Author Rebuttal · Authors · 2025-04-01
>
> We sincerely appreciate Reviewer bd1u's strongly positive feedback on our work. Your recognition of our clear writing, well-motivated approach, and the effectiveness of both our UniTraj dataset and TrajWorld architecture is truly encouraging. We are grateful for your support and share your belief that our work will contribute meaningfully to the broader scientific community.

---

### Official Review · Reviewer_8E9V · 2025-03-12

**Overall Recommendation:** 2

**Summary:**

This paper introduces the UniTraj dataset, which contains a large set of trajectories collected from 80 heterogeneous environments. It also presents a world model, TrajWorld, pretrained on this dataset. The pretrained world model demonstrates positive transferability to new environments in zero or few-shot settings. The paper primarily evaluates the trained world model in an off-policy evaluation setting for transition prediction policy evaluation.

**Claims And Evidence:**

The main claim is that by pretraining on a diverse set of trajectory environments, the pretrained world model can adapt to unseen environments with zero-shot or few-shot transfer, demonstrating improved transition prediction ability. Compared to previous models, the proposed model achieves better transferability in an offline setting. However, the authors did not test its ability to utilize the pretrained world model online, as was done in previous work with MPC.  There are also some questions for the experiment setup that need to be clarified.

**Essential References Not Discussed:**

The close related work is discussed Schubert et al. (2023), though regarding the novelty of the transformer architecture, factorized transformer [1] was not mentioned.

[1] Nayakanti, N., Al-Rfou, R., Zhou, A., Goel, K., Refaat, K.S. and Sapp, B., 2023, May. Wayformer: Motion forecasting via simple & efficient attention networks. In 2023 IEEE International Conference on Robotics and Automation (ICRA)(pp. 2980-2987). IEEE.

**Experimental Designs Or Analyses:**

After closely looking at the experimental design, I have the following concern regarding its setup.

Compared to Schubert et al. (2023), why did the authors choose to evaluate using off-policy evaluation rather than a more realistic setting where the learned WM is directly used to solve the task online through MPC in the real environment?

In the related work section, the main point authors is trying to argue is that Schubert et al. (2023) did not show a positive transfer in walker 2D environment but Schubert et al. (2023) evaluates differently compared to this work. I consider the evaluation setup of Schubert et al. (2023) is more challenging as they need to solve the problem with the real environment through MPC.

**Methods And Evaluation Criteria:**

Yes, for the most part, the proposed methods and evaluation criteria align with the problem and application at hand. The use of the UniTraj dataset for pretraining and the off-policy evaluation (OPE) setup are reasonable choices for assessing transferability in heterogeneous environments. However, there are some concerns relate to the experiment design which will be discussed in that section.

**Other Comments Or Suggestions:**

Figure 5 missing labels for each row.

**Other Strengths And Weaknesses:**

Strength:

The analysis in the experiment section is detailed and clear.
Proposed architecture performs better than the re-implementation of the baselines and demonstrate good transfer ability.

Weakness:
The proposed transform architecture look similar to the factorized attention in previous work, which makes the novelty a bit questionable.
The usefulness of the learned WM prior is questionable given the author did not use it online with the environment.

**Questions For Authors:**

In 5.2 the authors mention "Moreover, TDM predicts variants sequentially, which may accumulate errors and lead to less accurate results.", but TrajWorld also operates sequentially which can accumulate errors , could the author clarify?

TDM seems to produce a lot worse results than TW in Table 5, do the authors have any insights on this? Are the parameters count equivalent between the two models?

Could the author identify the major difference between the proposed interleaved temporal-variate attentions versus factorized attention introduced in Wayformer?

Did the author explore the different discretization strategy?

“Moreover, the transfer benefits are evident in both in-distribution and out-of-distribution scenarios.”  what does this sentence referring to?

Which policy would give you a more generalized world model, the expert or random policy or something in-between?

**Relation To Broader Scientific Literature:**

Modeling a world model is important for creating realistic simulations, especially when running such simulations is costly. This paper demonstrates that with a transformer-based world model and a diverse set of related environments, one can produce an adaptive world model prior.

**Theoretical Claims:**

N/A

---

> ### Author Rebuttal · Authors · 2025-04-01
>
> We sincerely thank Reviewer 8E9V for the thorough review and valuable questions.
>
> ## Q1: MPC evaluation
>
> Following Schubert et al., we have added **online MPC experiments**. In this setting, TrajWorld outperforms both baselines and its counterpart trained from scratch (see [anonymous figure](https://anonymous.4open.science/r/TrajWorld/mpc.pdf)). Due to space limitation, please refer to **response to W1 for Reviewer puvo** for details.
>
> **Discussions on transferability compared to Schubert et al.**: Given these new results, we can expand the discussions on Schubert et al.:
>
> > We demonstrate positive transfer to complex downstream environments such as Walker2D, not only for offline transition prediction/policy evaluation, but also for online MPC, which Schubert et al. did not. Our work differentiate from theirs in: (1) Setting: Instead of finetuning with $10^4$ episodes for MPC with random shooting, we more practically finetune with $10^2$ episodes for MPC with proposal policy; (2) Data diversity: Our UniTraj dataset emphasizes distribution diversity, rather than using pure expert trajectories; (3) Architecture: TrajWorld incorporates inductive biases tailored to the 2D structure of trajectory data for enhanced transferability. Notably, TDM exhibits negative transfer in our practical MPC setting. We believe our work complements and extends Schubert et al., offering new insights to the community.
>
> ## Q2: Difference with factorized attention in Wayformer
>
> We appreciate the feedback and acknowledge the relevance of Wayformer. We will include it as related work in the final version.
>
> While TrajWorld and Wayformer both adopt two-way attention (a.k.a. axial/factorized attention [1,2,3]) to handle a heterogeneous set of inputs with various numbers of dimensions, our work differs in several key aspects to preserve novelty:
>
> 1. **Tasks & architectures**: Wayformer targets motion forecasting, a regression task with heterogeneity in the multimodal **input** space. TrajWorld is designed for world modeling, an autoregressive task involving heterogeneity in both **input and output** space. This leads to significantly different macro and micro designs: **decoder-only with causal attention vs. encoder-decoder with bidirectional attention**.
> 2. **Homogeneous basics**: Wayformer handles various numbers of contextual objects across modalities via attention, but still relies on modality-specific projections to unify embedding dimensions. More thoroughly, TrajWorld achieves scalar-level homogeneity, **without the need for modality-specific projections** and is capable of zero-shot generalization to unseen input/output spaces.
> 3. **Transferability beyond efficiency**: Wayformer uses factorized attention mainly for efficiency, not showing a performance boost over full attention in its target task. In contrast, TrajWorld shows that inductive biases for the 2D structure **enhance transferability**, outperforming its 1D counterpart, TDM.
>
> To our knowledge, TrajWorld is the first to apply two-way attention in trajectory world modeling, and we hope it provides valuable insights to the community.
>
> [1] ViViT: A Video Vision Transformer.
>
> [2] Axial Attention in Multidimensional Transformers.
>
> [3] CCNet: Criss-Cross Attention for Semantic Segmentation.
>
> ## Q3: Other questions/clarifications
>
> - **Error accumulation in TDM**: TDM predicts sequentially along both variate and temporal dimensions due to its 1D architecture. TrajWorld, with its 2D architecture, predicts sequentially over time but jointly across variates at each timestep, which helps mitigate error accumulation.
> - **Worse results of TDM compared to TrajWorld**: As discussed above, we attribute TDM’s performance gap to error accumulation along sequences of variates and the lack of appropriate inductive biases for the 2D structure of trajectory data. Both models use the same parameter count.
> - **Discretization strategies**: Uniform discretization is widely used (e.g., in Gato, TDM, Farebrother et al.) and performs well in our experiments. Thus, we did not explore more complex methods like quantile-based binning.
> - **In- vs. out-of-distribution**: For transition prediction, we test trained models on both the same and different datasets. For instance, a model trained on hopper-medium-replay is tested on both hopper-medium-replay (in-distribution) and hopper-expert (out-of-distribution). Pre-training benefits TrajWorld in both scenarios.
> - **Which policy would give a more generalized world model**: We believe that diverse data from a range of policies (random, medium, expert, exploratory, etc.) yields a more generalizable model than trajectories from any single policy. We have conducted experiments pre-training our model on JAT, a subset of UniTraj with only expert data, which underperforms those pre-trained on the full data. Please refer to **response to Q1 for Reviewer 2GyG** for details.

---

> > ### Comment · Reviewer_8E9V · 2025-04-04
> >
> > I recommend the authors to make proper ablations on their propose architecture with the existing two-way attention architectures  if the authors consider the proposed architecture is a strong contribution to the paper.
> >
> > In addition, I recommend the authors to be clear about the statement on error accumulation in future revisions.

---

> > > ### Author Response · Authors · 2025-04-05
> > >
> > > We sincerely appreciate Reviewer 8E9V’s follow-up to our initial rebuttal. We aim to fully address your remaining concerns in this additional response.
> > >
> > > ### On Two-Way Attention Architectures
> > >
> > > In our work, we employ a two-way attention mechanism that interleaves attention across two data dimensions—timesteps and variates. This design shares the idea of using two-way attention in broader literature [1–5] (with different data dimensions). But our approach is tailored to the autoregressive trajectory world modeling setting, where the temporal attention is set to be causal.
> > >
> > > **Our architecture contribution lies in being first to introduce two-way attention into trajectory world modeling and demonstrating its benefits for transferability**. Existing two-way attention architectures like Wayformer [5] make valuable contributions in motion forecasting, but are not directly applicable to our trajectory world modeling task. On the other hand, the existing trajectory world model, TDM, does not utilize two-way attention and thus fails to fully exploit the inherent 2D structure of data. We provide comprehensive experimental results--both offline and online--comparing world models with and without two-way attention (TrajWorld vs. TDM), clearly demonstrating that introducing two-way attention significantly improves both performance and generalization in world model tasks.
> > >
> > > **Our architecture contribution does not claim to propose a new form of two-way attention or to benchmark the best use of two-way attention.** The two-way attention mechanisms used in prior works [1–4] are conceptually similar across different tasks, with variations primarily in application domains rather than in fundamental architectural design. We sincerely thank Reviewer 8E9V for highlighting Wayformer, which investigates two designs of two-way (factorized) attention: interleaved attention (with N/2 flips) and sequential attention (a single flip between dimensions). These approaches differ from our design, which performs N−1 flips. We explicitly do **not** claim that our interleaved scheme outperforms prior variants; rather, we believe this opens up a valuable direction for future exploration.
> > >
> > > As such, we respectfully believe that **the absence of detailed ablations against existing two-way attention forms does not diminish our contribution to the world model field**.
> > >
> > > ### Clarifying Our Broader Contribution
> > >
> > > Beyond architectural design, we want to highlight our significant contribution: **We investigate an under-explored world model pretraining paradigm across heterogeneous environments by integrating a newly-collected large-scale dataset, UniTraj, and a new world model architecture, TrajWorld**. The dataset and architecture are both designed based on our in-depth insights (see response to Reviewer 2GyG and 8E9V, respectively) into this particular but important setting. We conduct comprehensive experiments to **demonstrate--for the first time--positive transfer** from heterogeneous environments to complex downstream environments, while prior work, including Schubert et al., failed to achieve it. Our experiments span a variety of settings, including transition prediction, off-policy evaluation, and **model-predictive control per your advice**.
> > >
> > > **The key challenge of world models in the era of scaling is effectively leveraging all available trajectory data** from diverse, heterogeneous environments. Rather than making improvements within a mature setting, **our work opens up a way by proposing a systematic solution to this challenging problem**. We believe this represents a meaningful step toward establishing foundation world models capable of handling heterogeneous environments.
> > >
> > > ----
> > >
> > > Lastly, we appreciate your suggestion regarding clarifying error accumulation, and we will incorporate clearer explanations in future revisions.
> > >
> > > Your comments is taken very seriously. We hope this additional response clarifies the scope and significance of our contributions, and you can re-evaluate our work based on resolved misunderstandings.
> > >
> > > [1] ViViT: A Video Vision Transformer.
> > >
> > > [2] Axial Attention in Multidimensional Transformers.
> > >
> > > [3] MetNet: A Neural Weather Model for Precipitation Forecasting.
> > >
> > > [4] Axial-DeepLab: Stand-Alone Axial-Attention for Panoptic Segmentation.
> > >
> > > [5] Wayformer: Motion Forecasting via Simple & Efficient Attention Networks.

---

### Official Review · Reviewer_puvo · 2025-03-14

**Overall Recommendation:** 3

**Summary:**

The paper aims to tackle the heterogeneity issue in world model pretraining. To achieve this, the authors curate a unified trajectory dataset from 80 control environments. Based on this dataset, they introduce TrajWorld, a world model architecture that naturally accommodates varying sensors and actuators, thereby enabling efficient knowledge transfer across environments. The effectiveness of TrajWorld is validated across three unseen environments in terms of prediction error and policy evaluation reliability.

**Claims And Evidence:**

Yes, there are supported by clear and convincing evidence.

**Essential References Not Discussed:**

As far as I know, all closely related works are cited appropriately.

**Experimental Designs Or Analyses:**

The paper provides extensive experiments with sufficient details.

**Methods And Evaluation Criteria:**

The effectiveness of the proposed world model is assessed through prediction error and off-policy evaluation. However, an evaluation of model predictive control performance would be more convincing, as did by the closely related work TDM [1].

[1] A Generalist Dynamics Model for Control. Ingmar Schubert, et al.

**Other Comments Or Suggestions:**

A typo in Line 637: "the official repository repository" should be "the official repository".

**Other Strengths And Weaknesses:**

W1) **Missing MPC results.** While the proposed recipe improves zero-shot prediction and policy evaluation, its advantages are not demonstrated in model predictive control performance, which would provide an intuitive comparison to TDM.

W2) **Long-horizon comparison.** I agree that the sequential prediction of TDM may lead to error accumulation, but its flexible architecture could offer advantages for planning over long horizons. The paper does not specifically mention the experimental settings regarding horizons, so I am curious whether the proposed joint prediction scheme still dominates in long-horizon settings.

**Questions For Authors:**

Q1) **Scaling law.** As the authors have collected a large-scale dataset from 80 heterogeneous environments and demonstrated the benefits of pretraining, it would be interesting to see an analysis of the scaling laws regarding data diversity.

Q2) **Straight zero-shot prediction results.** The zero-shot prediction results in Figure 4 appear overly straight. Could you briefly discuss possible reasons for this phenomenon? Besides, I wonder if it is possible to visualize the predictions of other baselines for comparison? I believe it would be beneficial to demonstrate the advantages of TrajWorld.

**Relation To Broader Scientific Literature:**

The paper makes two key contributions in comparison to the related literature. First, it curates a large-scale trajectory datasets sourced from 80 control enviroments. Second, it presents a unified architecture to efficiently extract transferable knowledge from the heterogeneous data sources. These contributions could inspire future research on developing generalist world models.

**Theoretical Claims:**

I have checked the theoretical claims in this paper.

---

> ### Author Rebuttal · Authors · 2025-04-01
>
> We sincerely thank Reviewer puvo for the thorough review, insightful questions, and a positive evaluation of our work.
>
> ## W1: Model-predictive control (MPC) evaluation
>
> We have conducted MPC experiments comparing different world models.
>
> **Setup**: Following Schubert et al, we first attempted *MPC with random shooting planner*, but found it ineffective in our high-dimensional target environments, with any world models. We then adopt the *MPC with proposal* setting, also from Schubert et al. We set a practical scenario where world models trained on medium replay datasets are used to improve the medium-level trained proposal policies via MPC.
>
> **Implementation**: We use three medium-replay datasets from D4RL and medium-level policies from DOPE. The sample size is fixed at 128. The planning horizon is set to 25 steps for HalfCheetah and Walker, and 50 steps for Hopper to avoid myopic behavior in this fragile environment. Sampling noise is tuned for optimal performance across all world models: 0.05 (Hopper), 0.2 (Walker), 0.025 (HalfCheetah).
>
> **Results**: Our main results for MPC with proposal are shown in the table below and this [anonymous figure](https://anonymous.4open.science/r/TrajWorld/mpc.pdf). We find MPC improves proposal polices in Hopper and Walker but has little effect on HalfCheetah, which is more stable and less prone to failure. In contrast, Hopper and Walker are fragile, and the model helps prevent unsafe actions, leading to better planning.
>
> Overall, **MPC with TrajWorld delivers the best performing agents** compared to baseline models or its counterpart trained from scratch.
>
> |MPC w/ proposal & | MLP-Ensemble (w/o pt) | MLP-Ensemble (w/ pt) | TDM (w/o pt) | TDM (w/ pt) | TrajWorld (w/o pt) | TrajWorld (w/ pt) |*Proposal only*|
> |-|-|-|-|-|-|-|-|
> |Hopper|948±61|1091±125|1287±26|1117±145|1090±225|**1401**±236|*1078±143*|
> |Walker|3353±83|**3465**±20|3056±236|2619±36|2422±455|**3427**±370|*3049±104*|
> |HalfCheetah|5645±10|5692±19|5611±85|5647±25|**5858**±17|5809±15|*5697±30*|
>
> For MPC with random shooting, no world model provides successful agents. However, TrajWorld still performs relatively best among them (see [anonymous figure](https://anonymous.4open.science/r/TrajWorld/mpc.pdf)).
>
> **Efficiency**: TrajWorld predicts all variates jointly, unlike TDM which processes them sequentially. This leads to a major speedup: MPC for 1000 environment steps in HalfCheetah takes **40 minutes with TDM**, but only **3 minutes with TrajWorld**.
>
> We thank all reviewers for encouraging us to add these experiments, which further strengthen the contribution of our work. We will include them in the final version.
>
> ## W2: Long-horizon comparison
>
> We remark that our TrajWorld has a flexible architecture similar to TDM and can also handle arbitrary prediction horizons (only constrained by its training context length).
>
> Below, we summarize our experimental settings regarding horizons:
>
> - **Zero-shot generalization** (Fig. 4b): rollouts for **10 steps**. We have expanded the baseline results for comparison (see Q2 below).
> - **Transition prediction** (Sec. 5.2): short horizon, report **one-step** prediction error.
> - **Off-policy evaluation** (Sec. 5.3) involves rollouts of an extremely long horizon (**2000 steps** as mentioned in App B.4.1). Due to our model's short context length of 20 (limited by our computational resources), this is done by sliding windows.
> - **Model-predictive control**: rollouts over relatively long horizons of 25 or 50 steps.
>
> Across all these scenarios, TrajWorld provides superior performance compared to baselines.
>
> ## Q1: Scaling laws regarding data sizes and diversity
>
> We have conducted an analysis on the effects of different data sizes and diversity for pre-training. Our results in this [anonymous figure](https://anonymous.4open.science/r/TrajWorld/data-analysis.pdf) validate that both the large scale and diversity contribute to the effectiveness of pre-training.
>
> Due to space limitation, we kindly refer the reviewer to **response to Q1 for Reviewer 2GyG** for experimental details and results.
>
> ## Q2: Straight zero-shot prediction
>
> We suppose that the overly straight predictions are likely due to the short context window (10 steps) and slow motion speed. Through this context, the zero-shot model is not able to precisely capture the quadratic relationship between state and action, and is only able to reflect coarse directional changes.
>
> For comparison, we also provide zero-shot prediction from other baselines in this [anonymous figure](https://anonymous.4open.science/r/TrajWorld/zero-shot-prediction.pdf). As shown, in an unseen environment, both TDM and MLP baselines fail to generalize, producing incorrect predictions and failing to capture the underlying state-action relationship at all. Specifically, TDM fails to predict how push forces from two opposite directions lead to different x positions. On the other hand, MLP fails to produce any reasonable results with extreme error accumulation.

---

> > ### Comment · Reviewer_puvo · 2025-04-09
> >
> > Thank the authors for their efforts to address my concerns. I will keep my initial rating.

---

> > > ### Author Response · Authors · 2025-04-09
> > >
> > > We sincerely appreciate the time you took to read our rebuttal and engage with our responses. We're glad we can address your concerns, and your positive assessment is a great support for us to believe in the value of our work.

---

### Official Review · Reviewer_2GyG · 2025-03-15

**Overall Recommendation:** 3

**Summary:**

This paper presents a trajectory world model that handles varying sensor and actuator information across different environments. To support the generalization of the world model, this work composes a large dataset, UniTraj, comprising over one million trajectories from 80 environments. The key ingredient of the proposed method is to model the dynamic transitions on discretized variates over the temporal horizon. The learned world model demonstrates effective positive transfer across heterogeneous and complex control environments while achieving a new state-of-art for off-policy evaluation.

**Claims And Evidence:**

Yes, the claims made in this submission are supported by clear and convincing evidence.

**Essential References Not Discussed:**

N/A.

**Experimental Designs Or Analyses:**

Issues of experimental designs:


- **Lack of experiments on decision-making**. The paper did provide one means of utilizing the pre-trained world model, which is off-policy evaluation in Sec.5.3.  However, investigations on using this model to improve policy through reinforcement learning or sampling-based optimizations are underexplored.

**Methods And Evaluation Criteria:**

Issues of methods:

- **Limited contribution regarding the UniTraj dataset**. This paper introduces the UniTraj dataset by composing large-scale data from different environments. However, there are limited contributions in this regard. First, the dataset seems to be a simple composition without any rigorous curation, selection, or filtering process. Further, even though the paper demonstrated that using UniTraj in pre-training can improve downstream performances as shown in Fig.1 and Fig.5, there is a lack of critical studies on how different data scales and diversity affect model performance. For instance, how would the world model generalize when pre-trained on a subset of UniTraj, such as 1/2 or 1/10? Given these issues, the introduction of this dataset cannot form a valid technical contribution.


- **Limited novelty of the proposed TrajWorld architecture compared to TDM**. According to Fig.3 and descriptions in L322-324, the main difference between TDM[1] and the proposed TrajWorld is the newly introduced temporal dimension. However, leveraging the temporal dimension with additional attention modules has been widely studied in the literature on world models[2] and video generation models[3, 4]. Such an adaptation does not bring further insights to the community.

[1] A Generalist Dynamics Model for Control

[2] Generalized Predictive Model for Autonomous Driving

[3] AnimateDiff: Animate Your Personalized Text-to-Image Diffusion Models without Specific Tuning

[4] Align your Latents: High-Resolution Video Synthesis with Latent Diffusion Models

**Other Comments Or Suggestions:**

N/A.

**Other Strengths And Weaknesses:**

Strengths:
- The paper is well-organized and easy to follow
- Experiments on how the pre-training influences downstream fine-tuning are meaningful. As shown in Fig.5, the authors conducted a wide range of experiments to demonstrate that large-scale pre-training is helpful for downstream fine-tuning with various train-test pairs.
- The idea of unifying world modeling on heterogeneous environments through in-context variates is intriguing and worth exploring.

Other weakness:
See weaknesses in the above sections.

**Questions For Authors:**

N/A.

**Relation To Broader Scientific Literature:**

The key contributions of the paper relate to the areas of world models, reinforcement learning, and pre-training on large-scale dataset.

**Theoretical Claims:**

No issues with the theoretical aspect of this work.

---

> ### Author Rebuttal · Authors · 2025-04-01
>
> We sincerely thank Reviewer 2GyG for the thoughtful review and valuable comments, especially the recognition of our idea of unifying world modeling across heterogeneous environments.
>
> ## Q1: Dataset contribution
>
> **Dataset construction**: We respectfully disagree with the assessment that our dataset is a simple composition. We elaborate on the insights behind UniTraj as follows:
>
> 1. **Selection**: We carefully **selected** the subsets to assemble UniTraj. Unlike Schubert et al. (2023), which utilizes only expert or near-expert trajectories, our dataset emphasizes distribution diversity beyond sole environment diversity, as detailed in Sec 3.
> 2. **Additional collection**: To further enrich environment diversity, we **collected** new trajectories ourselves from Modular RL, going beyond existing datasets.
> 3. **Filtering**: As noted on Line 124, we **filtered** all trajectories from three downstream environments, contributing a reasonable testbed for cross-environment transfer to the community.
> 4. **Weighting**: We manually **weighted** different subsets, trying to balance size and diversity. For example, DB-1 is oversampled due to its too small size. We apologize for not including these weights in the appendix and provide them below:
>
> | Subsets                        | ExoRL | RLU | JAT | DB-1 | TD-MPC2 | Modular RL |
> | ------------------------------ | ----- | --- | --- | ---- | ------- | ---------- |
> | (Unnormalized) sampling weight | 75    | 5   | 90  | 1    | 90      | 30         |
>
> **Analysis of dataset scales and diversity**: We appreciate the suggestion to analyze the impact of pre-training data. We conducted new experiments by pre-training three versions of TrajWorld on different subsets of UniTraj—namely, 1/10 size, 1/100 size, and the JAT subset (with only expert trajectories from 5 environments)—followed by fine-tuning on downstream tasks.
>
> For **transition prediction**, we adopt a challenging setting: for each environment, we train models on the expert dataset, and test them on datasets of all levels. We also provide results for **model-predictive control** (see experimental details below).
>
> The results, shown in this [anonymous figure](https://anonymous.4open.science/r/TrajWorld/data-analysis.pdf), compare these new models with one pre-trained on the full UniTraj and another trained from scratch. We observe that all subset pre-trained models underperform the fully pre-trained one, revealing a scaling law with respect to data size. These findings underscore the **importance of both scale and diversity** in pre-training data, and strengthen our contribution in advocating for large-scale, heterogeneous pre-training and in constructing the UniTraj dataset to support it.
>
> ## Q2: Architecture novelty
>
> We respectfully believe there may have been a misunderstanding regarding our architectural contributions in comparison to TDM:
>
> - TDM actually models tokens across both the variate and temporal dimensions, similar to ours. But it **flattens them into a one-dimensional sequence** and applies the original GPT architecture. This approach discards the inherent 2D structure of trajectory data and may 'not bring further insights to the community'.
> - Compared to TDM, our TrajWorld does not 'newly introduce the temporal dimension', but rather **preserves and exploits the natural 2D structure** by employing a two-way attention mechanism, with each one capturing relationships within its respective dimension.
>
> The superior performance of TrajWorld underscores the importance of appropriate inductive biases for enhancing transferability in trajectory world models.
>
> ## Q3: Experiments on decision-making (MPC)
>
> We have added experiments on improving policies via sampling-based optimization (**model-predictive control**). In this setting, **TrajWorld outperforms both baseline models and its counterpart trained from scratch** (see this [anonymous figure](https://anonymous.4open.science/r/TrajWorld/mpc.pdf)).
>
> Due to space limitation, we kindly refer the reviewer to **response to W1 for Reviewer puvo** for detailed experimental setup and results.

---

> > ### Comment · Reviewer_2GyG · 2025-04-07
> >
> > Thanks for the detailed response. The additional ablation study on data scales and the experiments on decision-making are convincing to me. Therefore, I'd like to increase my score to 3. Weak accept (i.e., leaning towards accept, but could also be rejected).

---

> > > ### Author Response · Authors · 2025-04-08
> > >
> > > Dear Reviewer 2GyG,
> > >
> > > We sincerely appreciate your thoughtful re-evaluation of our paper and the subsequent positive rating. Your constructive feedback, particularly regarding the ablation study on data scaling and MPC experiments, has been invaluable in helping us improve the work.
> > >
> > > Thank you again for your insightful feedback.
> > >
> > > Best regards,
> > >
> > > Authors

---

### Decision · Program_Chairs · 2025-05-01

**Decision:**

Accept (poster)

**Comment:**

This paper proposes UniTraj, a dataset of many offline RL tasks, and TrajWorld, a world model that can train on all of them.

First the dataset, it seems to be a concat of many existing datasets. The world model (TrajWorld) is a fairly off the shelf architecture, with a spatiotemporal factorization like is done in many video models circa 2022-2024.

The overall premise is promising, "building a foundation world model", yet, the approach doesn't quite live up to it. It is quite likely that to really do this would require leveraging internet videos, and it seems very unlikely we would want a foundation model for small scale proprioceptive tasks.